# Accelerating Deep Learning with Dynamic Data Pruning

## Abstract

Deep learning's success has been partially attributed to the training of large, overparameterized models on massive amounts of data. As this trend continues, model training has become prohibitively costly, requiring access to powerful computing systems to train state-of-the-art networks. A large body of research has been devoted to addressing the cost per iteration of training through various model compression techniques like pruning and quantization. Less effort has been spent targeting the number of iterations. Previous work, such as forget scores and GraNd/EL2N scores, address this problem by identifying important samples within a full dataset and pruning the remaining samples, thereby reducing the iterations per epoch. Though these methods decrease the training time, they use expensive static scoring algorithms prior to training. When accounting for the scoring mechanism, the total run time is often increased. In this work, we address this shortcoming with dynamic data pruning algorithms. Surprisingly, we find that uniform random dynamic pruning can outperform the prior work at aggressive pruning rates. We attribute this to the existence of "sometimes" samples—points that are important to the learned decision boundary only during some of the training time. To better exploit the subtlety of sometimes samples, we propose two algorithms to dynamically prune samples and achieve even higher accuracy than the random dynamic method. We test all our methods against a full-dataset baseline and the prior work on CIFAR-10, CIFAR-100, CINIC-10, and ImageNet, and we can reduce the training time by up to $10\times$ without significant performance loss. Our results suggest that data pruning should be understood as a dynamic process that is closely tied to a model's training trajectory, instead of a static step based solely on the dataset alone.

## 1 Introduction

A growing body of literature recognizes the immense scale of modern deep learning (DL) (Bommasani et al., 2021; Strubell et al., 2019), both in model complexity and dataset size. The DL training paradigm utilizes clusters of GPUs and special accelerators for days or weeks at a time. This trend deters independent researchers from applying state-of-the-art techniques to novel datasets and applications, and even large research organizations accept this approach at significant costs.

Currently accepted methods (Narang et al., 2018) target the per-iteration computational cost of the model during training; however, not as much effort has been spent on reducing the total number of training iterations. Since even simple datasets (Krizhevsky, 2009) require hundreds of epochs over tens of thousands of samples, eliminating a non-essential subset of data presents a promising opportunity for efficiency.

Work from other DL domains suggest that only a subset of the data influences the decision boundary and contributes a majority of the loss incurred by the model (Agarwal et al., 2021; Toneva et al., 2019). Furthermore, curriculum learning (Bengio et al., 2009) asserts that samples can be ranked which might allow us to prune redundant "easy" samples. Prior work on data pruning (Toneva et al., 2019; Paul et al., 2021) take advantage of this property to eliminate a majority of the dataset without incurring significant performance loss. Unfortunately, these methods run their scoring algorithm prior to training and require one or more passes over the dataset. When we include the cost of scoring the samples, the total run time exceeds the time it takes to do a single conventional training run. This prevents researchers from utilizing the prior work on new, non-standard datasets.

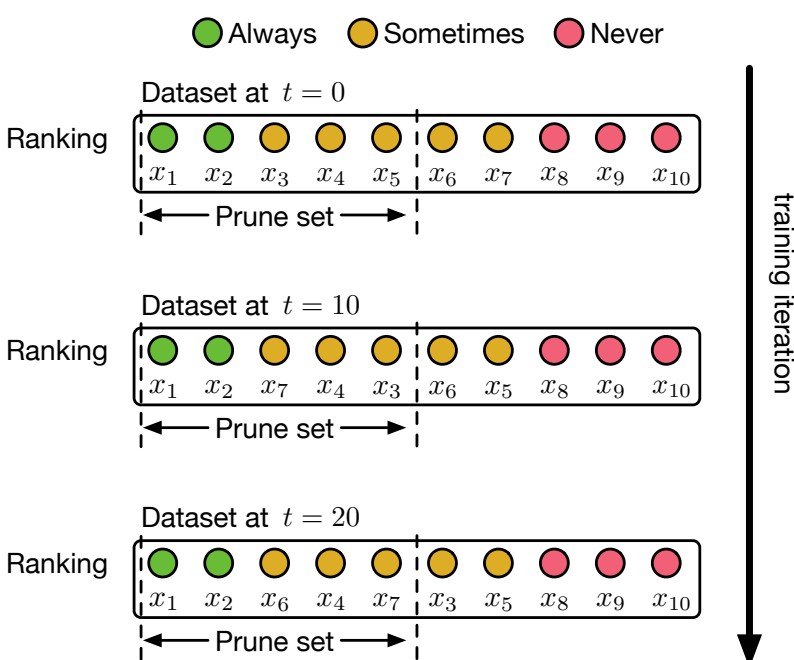

Figure 1: Dynamic data pruning presents multiple opportunities for a sample to be selected for training. This illustration shows our finding that datasets are separated into three groups—samples that are always selected, samples that are never selected, and samples that are selected only some of the time. Static methods fail to effectively target the last group, since their ranking varies during training.

In our work, we dynamically select a subset of the data at fixed checkpoints throughout training. We make the following new observations and contributions:

1. By measuring the loss distribution over samples, we find that a dataset can be qualitatively split into three groups—always samples, never samples, and sometimes samples (see Fig. 1). Always samples are selected at nearly every scoring checkpoint. Similarly, never samples are rarely selected. Sometimes samples are selected at only some checkpoints, and their inclusion is highly variable across different training runs. Static pruning methods, like the prior work, can identify always or never samples but fail to effectively target sometimes samples. In fact, we find that randomly selecting a subset of the data at each checkpoint is more effective than the static baselines.

2. To better target sometimes samples, we design a dynamic scoring mechanism based on per-sample loss. Despite scoring more frequently, our mechanism reduces the total run time including the cost of scoring, while the prior work typically increases it. Moreover, at aggressive pruning rates, we obtain higher final test accuracies on CIFAR-10, CIFAR-100, CINIC-10, ImageNet, and a synthetically modified variant of CIFAR-10.

3. Since the sometimes samples vary in importance across training runs, we note that the optimal dynamic scoring selection is tightly coupled to the model training trajectory. So, we re-frame the data pruning problem as a decision making process. Under this lens, we propose two variants of our scoring mechanism based on commmonly known $\epsilon$-greedy and upper confidence bound (UCB) algorithms. With these additional improvements, we obtain even higher performance at aggressive pruning rates even when the dataset is imbalanced.

## 2 Related Work

### 2.1 Curriculum learning, active learning, and coresets

The idea that certain samples are more influential than others has been observed for some time in curriculum learning and active learning. Curriculum learning exploits this to present samples to the model under a favorable schedule during training to obtain better generalization (Bengio et al., 2009; Cai et al., 2018; Kumar et al., 2010). Our work has a similar objective, but we are concerned with ranking samples in order to eliminate them instead of schedule them. Furthermore, the notion of "easy samples" from curriculum learning is not directly equivalent to "important samples" in data pruning. For example, one could imagine that a sample far from decision boundary but correctly classified is easy, but it is not informative and a wasted use of the pruning sample budget. Nonetheless, curriculum learning and our work both use the loss as a proxy for scoring samples.

Active learning attempts to rank new, un-labeled samples based on redundancy instead of difficulty. Samples are selected via an acquisition function, mirroring the selection criterion in our work. One of the computationally cheap techniques in this setting is uncertainty sampling (Settles (2009)), which we use in our work as well. It is possible that more sophisticated acquisition functions (e.g. BatchBALD (Kirsch, 2019)) could lead to better performance that our methods, but their complexity incurs a run-time penalty. Thus, in our context, where the cost of scoring is critical, any additional overhead must be thoroughly justified. This underscored by our findings in Sec. 4.2 that random sample selection is high performing for many datasets.

Another line of related work, coresets are primarily concerned with finding a subset of the data that can approximate the entire dataset Sener & Savarese (2018); Settles (2009); Tsang et al. (2005); Munteanu & Schwiegelshohn (2018); Munteanu et al. (2018); Campbell & Broderick (2018). Some of these approaches come with guarantees that the coreset approximates the full dataset, but they are difficult to scale up to deep networks. More importantly, while there might be strong overlap between samples in a coreset and always samples in our work, we find that sometimes samples make a non-negligible contribution to the accuracy. Unlike always samples, sometimes samples need to rotate in and out of the pruning subset, so it is unclear how a coreset framework could correctly identify them.

### 2.2 Static data pruning

Several works have studied how to statically prune a dataset prior to training. Toneva et al. (2019) proposed the concept of forget scores, which track the number of times an example transitions from classified correctly to misclassified throughout training. They classified the data into two sets—forgettable examples (frequently misclassified) and unforgettable examples (rarely misclassified). By pruning samples based on the forget scores, they eliminate up to 30% of the data with no loss in performance on CIFAR-10. An important limitation of their method is the need to perform a full training run to obtain the forget scores. Coleman et al. (2020) build on forget scores in their work called selection via proxy (SVP). SVP uses a smaller proxy model to rank examples in order to choose a subset that can be used to train a larger model. By re-using the forget scores algorithm and training a smaller network for fewer epochs, they obtain a subset of samples to train a larger model with no loss in performance even when the subset is 50% of the original dataset size. Unlike the forget score-based methods, Paul et al. (2021) use the gradient norm-based (GraNd) scores of individual samples averaged across 10 different random models to prune data at initialization. They also prune up to 50% of CIFAR-10 with no loss in accuracy. Even though the SVP and GraNd methods require less computation than the original forget score work to obtain a ranking, the results are too noisy to statically prune the data. Thus, they must rely on more iterations or averaging across trials to obtain the final subset of samples. As a result, all three methods incur a high overhead for pruning that increases the total run time to train the final model. This prevents these methods from being applied to new datasets for the first time. Since our method is dynamic, it can be applied directly to new data without *a priori* information. Furthermore, our dynamic approach allows us to exploit sometimes samples, resulting in more aggressive pruning rates.

## 2.3 Other online selection methods

We are not the first work to consider selecting samples in a dynamic fashion. Prior work on online batch selection (Loshchilov & Hutter, 2016) and deep learning with importance sampling (Katharopoulos & Fleuret, 2018) also attempt to accelerate training by dynamically selecting subsets of samples. The methods proposed in those works are more complex, and this is justified based on the assumption that samples are neatly ordered. A key finding of this paper is that most samples in a dataset have no distinct ordering, so many of the additional heuristics built into the prior work might be superfluous. This is an important consideration when more complex methods lead to lower overall training speedup. Moreover, while the prior work relies on sometimes samples to attain better performance, this work is the first to identify and characterize their existence. It would be interesting to study how sometimes samples influence these prior methods, but this is out of the scope of this paper.

# 3 Methods

In this section, we describe our approach to data pruning. First, we re-frame data pruning as a dynamic decision making process. Next, we present our scoring mechanism based on filtered uncertainty sampling. Finally, we discuss how the $\epsilon$-greedy and upper confidence bound algorithms can be used on top of our scoring mechanism to further improve the final test accuracy.

## 3.1 Dynamic data pruning

Static data pruning assumes that the optimal subset of samples depends primarily on the data distribution. Model training is a means to an end, and the training trajectory of a specific model does not directly factor into the final pruned dataset. In our work, instead of approaching data pruning as a one-time step, we view it as a dynamic process that is coupled to a given model's training run. The total number of training epochs is divided by fixed pruning checkpoints. At each checkpoint, an algorithm may observe the current performance of the model on the full dataset. Given this observation and information retained from previous checkpoints, the algorithm makes a decision on which points to use for training until the next checkpoint. This process is described in Algorithm 1.

---
**Algorithm 1** Dynamic data pruning process

---
**Require:** $k$ = # of samples to keep, $s$ = pruning selection criterion, $T_p$ = pruning period, $T$ = total # of epochs, dataset $X$ (w/ $N$ samples), classifier $f_\theta$
  **for** $T/T_p$ passes **do**
    Select subset, $X_p = s(f_\theta, X, k)$
    **for** $T_p$ epochs **do**
      Train $f_\theta$ on $X_p$
    **end for**
  **end for**

---

## 3.2 Pruning selection criterion

In the next subsections, we describe the different selection criteria, $s$, that are proposed in our work.

### 3.2.1 Uncertainty sampling with EMA

Uncertainty sampling preferentially selects samples for which the model's current prediction is least confident (Settles, 2009). In general, it is quick to compute and identifies points close to the estimated decision boundary which is a measure of the informativeness of a sample. Specifically, we use the per-sample cross entropy loss for assigning a score to individual sample $x$ with label $y$ over $C$ classes:

$$s_{\text{uncertainty}}(x) = \sum_{i=1}^{C} -y_i \log \left( \frac{\exp(f_\theta(x_i))}{\sum_{j=1}^{C} \exp(f_\theta(x_j))} \right) \tag{1}$$

This scoring criterion is similar to the EL2N score from Paul et al. (2021). In that work, the final EL2N score is obtained by averaging the uncertainty over several trials to reduce the noise given in a single trial. We observe the same noisy behavior across pruning checkpoints in our work. A network's decision boundary can vary dramatically, especially with respect to points which are included in one checkpoint and excluded in another. Since we are pruning dynamically, instead of a static average, we use an exponential moving average (EMA) filter:

$$s_{\text{ema}}(x) = \alpha s_{\text{uncertainty}}(x) + (1 - \alpha)s_{\text{previous ema}}(x) \tag{2}$$

where $s_{\text{previous ema}}(x)$ is the value of $s_{\text{ema}}(x)$ from the previous checkpoint and $\alpha$ is the constant linearly interpolating between two quantities. We select the $k$ samples with highest score.

### 3.2.2 $\epsilon$-greedy approach

So far, our decision making criterion has been fairly simple. The points with the highest average loss are selected at each checkpoint. This approach can be viewed as a greedy solution to the decision making process described previously. But our estimate of the uncertainty of each point is noisy, and since we maximally exploit our current information, we might under-select some points and obtain a suboptimal solution. A simple fix is to encourage more exploration by randomly selecting points which do not currently have the highest score. This is referred to as an $\epsilon$-greedy approach (Sutton & Barto, 2018).

Concretely, at every frequency checkpoint, $(1 - \epsilon)k$ points are selected using Eq. (2), and $\epsilon k$ points are selected from the remaining data uniformly at random. Our results show that this additional exploration can help boost performance even further.

### 3.2.3 Upper-confidence bound approach

While the $\epsilon$-greedy approach does introduce more exploration, it does so in a random manner. A more directed approach would be to prioritize samples for which the running variance of Eq. (1) is high even when the mean is low. In this way, when we do explore, we choose to focus on the samples who's scores are poorly estimated.

The upper-confidence bound (UCB) (Brochu et al., 2010) algorithm is another method that selects samples based on the mean value and the variance. Assuming the initial mean scores are aggregated from the random model and the starting variance is zero, we compute the variance based on Welford's algorithm (Welford, 1962):

$$\begin{aligned} \text{var}(x) = &(1 - \alpha)\text{var}_{\text{previous}}(x) \\ &+ \alpha(s_{\text{uncertainty}}(x) - s_{\text{previous ema}}(x))^2 \end{aligned} \tag{3}$$

With the mean and variance computed, the final score of the sample is computed by adding the two quantities together:

$$s_{\text{ucb}}(x) = s_{\text{ema}}(x) + c\text{var}(x) \tag{4}$$

where $c$ is a hyper-parameter which dictates the the variance's influence on the total score.

## 4 Results and discussion

We experimentally evaluate our methods on the CIFAR-10 and CIFAR-100 datasets (Krizhevsky, 2009) following the setup in Paul et al. (2021). Each image is augmented with standard transformations such as normalizing the images by per channel mean and standard deviation, padding by 4 pixels, random cropping to 32 by 32 pixels, and horizontal flipping with probability 0.5. We use ResNet-18 (He et al., 2016) with

the initial 7x7 convolutional layer and 2x2 pooling layer swapped out for a 3x3 convolution layer and a 1x1 pooling layer to account for change in image resolution. We train our model for 200 epochs with the stochastic gradient descent (SGD) optimizer with an initial learning rate = 0.1 with Nesterov momentum = 0.9, weight decay = 0.0005, and batch size = 128. The learning rate was decayed by a factor of 5 at 60, 120, and 160 epochs. Our implementation is based on PyTorch (Paszke et al., 2017) and all experiments were conducted on a 11GB NVIDIA GeForce RTX 2080 Ti GPUs. For CIFAR100, we use the same experimental setup but swap out ResNet-18 for ResNet-34.

For our baselines, we compare against conventional training with the full dataset, forget scores, and EL2N scores. For the forget scores, we train the model for 200 epochs and compute the forget scores as per the prior work. For the EL2N scores, we also follow the scoring algorithm laid out in the prior work and average over 10 independent models trained for 20 epochs each. After obtaining the forget and EL2N scores, we statically prune the data set by selecting the top scoring samples based on the designated pruning rate. For all our experiments, we run each method over 4 independent random seeds. We also present results on the distribution of selected samples to corroborate our findings on sometimes samples. For these results, we use the Turing package (Ge et al., 2018) in the Julia language (Bezanson et al., 2017) to fit a mixture model with Bayesian MCMC methods to delineate between sample groups. For all distributions, we use the Metropolis-Hastings algorithm (Robert, 2016) with 1000 samples and 3 chains (averaged).

## 4.1 CIFAR-10 results

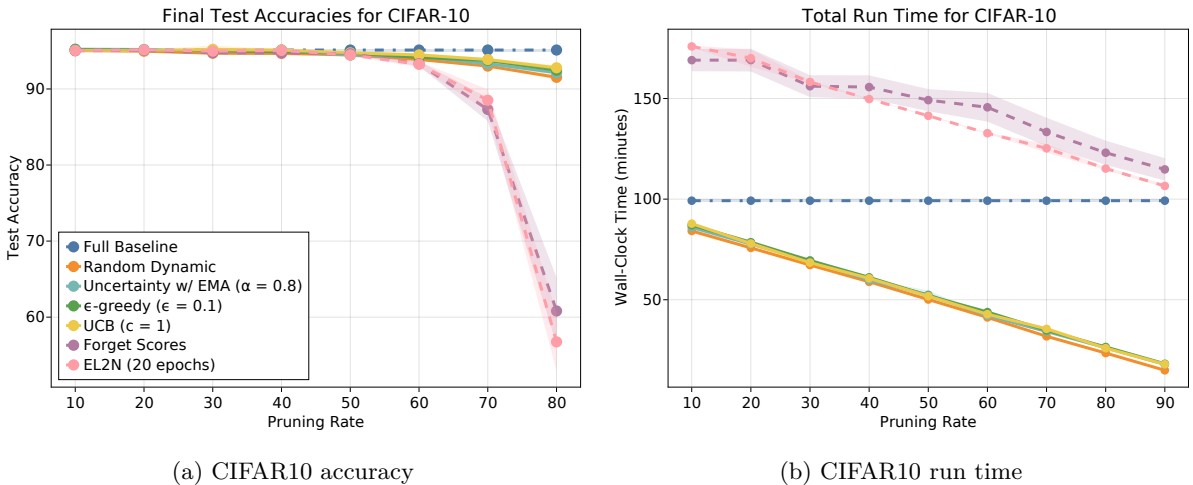

(a) CIFAR10 accuracy           (b) CIFAR10 run time

Figure 2: Dynamic data pruning applied to CIFAR-10 with ResNet-18. Fig. 2a shows the final test accuracies for each method and Fig. 2b shows the run time to execute the approach (including scoring cost). Zoomed version available in supplementary material.

Fig. 2 shows the result of dynamic data pruning with the various selection methods introduced in the prior section. We sweep the pruning rate with the pruning period ($T_p$) set to every 10 epochs. For all methods that use an EMA, we set $\alpha = 0.8$.

Below 50% of the data pruned, all methods appear to perform equally well. At more aggressive rates, the static baselines drop dramatically in performance. Our proposed methods continue to maintain reasonable performance even when 80% of the dataset is pruned. As expected, the $\epsilon$-greedy and UCB improve the performance slightly over the uncertainty with the EMA.

In Fig. 2b, we plot the measured wall-clock time for each method. Since the static methods perform pruning prior to training with an overhead equivalent to training on the full data, their wall-clock time is nearly double the full baseline (when omitting the static overhead, the wall-clock time is on par with random pruning). In contrast, all the dynamic methods reduce the wall-clock time even at modest pruning rates.

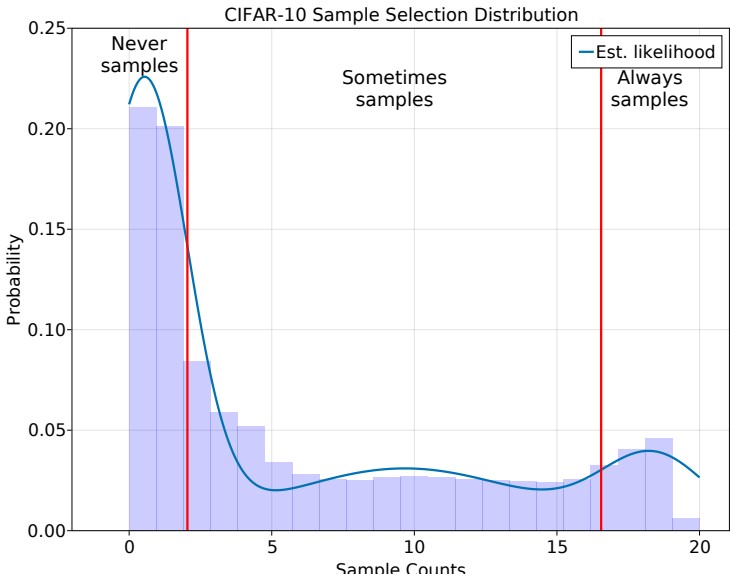

Figure 3: The density function (of a mixture model) of sample counts from 4 independent CIFAR-10 runs during our uncertainty with EMA method at 70% pruning. The horizontal axis represents the number of times a sample has been counted by the selection process. The data is separated into three groups—samples that are always selected, samples that are selected only some of the time, and samples that are never selected. The red lines are denote one standard deviation from the mean of the left and right mixture components. Static methods fail to effectively target the middle group.

## 4.2 Unreasonable effectiveness of random pruning

The most surprising result in Fig. 2a is the performance of the random dynamic pruning algorithm. This method randomly samples the training data at each checkpoint without any consideration to the uncertainty or previous checkpoints. Still, the approach is able to outperform the static baselines, and it suffers only a 3.5% drop in accuracy at a pruning rate of 80%. While this is unexpected in the context of this paper, it does match intuition from first principles. The very basis of SGD training relies on the ability to train on randomly sampled batches. If there was a strong bias in the relative importance of samples, then most DL training would be quite inefficient.

This is corroborated by our findings on the existence of "sometimes" points. When performing each of the trials to obtain Fig. 2, we count the number of times each sample is selected across all checkpoints and trials. In Fig. 3, we plot the normalized histogram of these counts. We fit a mixture of three truncated normal distributions (between zero and the total number of checkpoints) using the Turing probabilistic programming language as mentioned at the start of this section. The resulting density function (likelihood) is plotted on top of the histogram in Fig. 3.

We observe that the curve can be separated into three distinct regions—always, sometimes, and never samples. Always samples are selected at nearly every checkpoint; moreover, the standard deviation in this region is low based on the range of occupied sample counts and variance of the right Gaussian component after the mixture model fit was completed. Similarly, the never samples are rarely selected, based on the left Gaussian component occupying a small number of sample counts and a tight variance. Like always samples, these points are consistently never selected.

Contrary to always and never samples, sometimes samples are not consistently selected or omitted. The middle mixture component fits a large section of the support. This wide range of sample counts indicates the high variance in this region and shows that the ranking of sometimes samples is not consistent. More

importantly, a significant portion of the mass is contained in this component, suggesting that a large subset of the dataset cannot be classified as always or never selected.

Static pruning methods largely target the elimination of never samples. By selecting the top $k$ points in a single pass, they successfully eliminate never samples and keep always samples; however, the small mass within the always region means that static methods must select some arbitrary subset of sometimes samples. In contrast, dynamic methods can rotate their selection of sometimes samples based on which ones are most beneficial to a specific model's training trajectory. Most importantly, since such a large fraction of the dataset does not have a consistent ranking, even randomly sampling these points (as the random dynamic method does) is better than statically committing to a fixed subset of them.

Table 1: The effect of randomly selecting sometimes samples when re-training a model using previous scores.

| Method | Accuracy |
|---|---|
| Original dynamic training | 93.33% |
| Always set + static sometimes set | 89.52% |
| Always set + random sometimes set | 92.50% |

We test this hypothesis by training a model on CIFAR-10 using the uncertainty with EMA method at a pruning rate of 70%, and we save the scores across checkpoints. Then, we train a new initialization with dynamic pruning using two different subsets of the original dataset. First, we choose the top 30% of samples according to the scores from the final checkpoint. This creates a static policy that uses the same subset at every checkpoint. Second, we choose only the always samples, which make up about 15% of the data, and we choose the remaining samples in the pruning budget uniformly at random for each checkpoint. The difference is shown in Tab. 1. Statically choosing the sometimes samples results nearly a 4% performance drop. Dynamically sampling from the remaining samples only results in a 1% performance degradation.

### 4.3 Training with fewer epochs

We have shown in Section 4.1 that dynamic data pruning is effective on CIFAR10. One may wonder if the random dynamic method's performance is high simply due to the fact that the baseline was trained for 200 epochs. That is, CIFAR10 is a comparatively simple dataset for a model like ResNet18, and if the number of epochs was increased/inflated, then even random subsets of the data approach the performance of the baseline trained on the full dataset. This assumption leads to two lines of inquiry: first, are our methods robust to reduced number of epochs, and second, is random dynamic data pruning equivalent early stopping?

To verify if the number of epochs has an impact on the performance of the dynamic methods, we repeat the experiments in Section 4.1 except for 100 epochs while scaling the learning rate scheduler to match the change in epochs. Fig. 4 shows the results of this experiment. By comparing the accuracy of the full baseline with 100 to 200 epochs, we observe a difference of 0.72% across 4 different random seeds, which shows that the additional epochs of training provide small but nontrivial benefit. When we analyze the static methods, we see that they behave similarly to the 200-epoch case as they drop off rapidly in performance as we see in Sec. 4.3. The random dynamic method at a pruning rate of 70% maintains a margin of roughly 2.5% from the full baseline compared to a margin of 2% for CIFAR10 training with 200 epochs. The other dynamic methods including the uncertainty with EMA, $\epsilon$-greedy and UCB are able to maintain the same trend as with 200 epochs until 80% pruning before uncertainty with EMA and $\epsilon$-greedy drop in performance. UCB, however, still remains higher than the random dynamic method. From the result of the experiment, we can say that training for more epochs is certainly beneficial for dynamic data pruning in terms of accuracy but our overall conclusions do not change that dynamic data pruning enables more aggressive pruning of the dataset than static pruning.

To investigate the relationship between data pruning and early stopping, we focus on the "samples seen" by the model, i.e.

$$\text{samples seen} = kN \times T \tag{5}$$

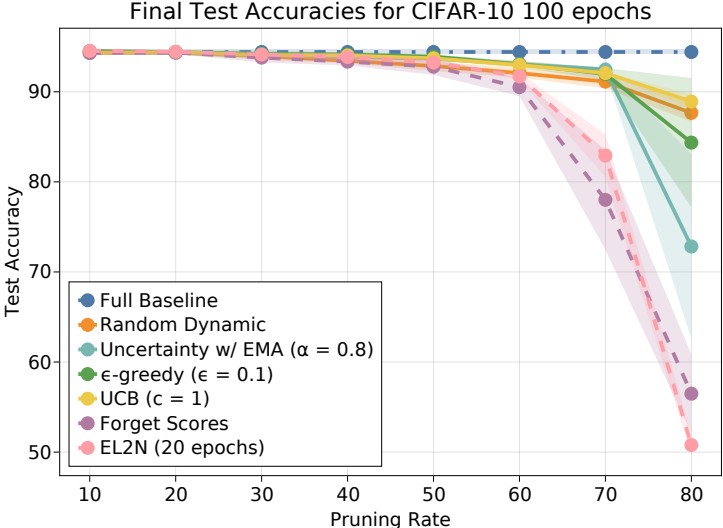

Figure 4: Dynamic data pruning CIFAR-10 with ResNet-18 with 100 epochs, showing the final test accuracy for each method. With a truncated training time, the separation between various methods increases, but the main conclusions of the paper remain the same.

We can accelerate training by reducing the samples seen per epoch, $kN$, or we can reduce the number of epochs, $T$. The former corresponds to data pruning while the latter is early stopping. Qualitatively, there is a difference between these two approaches. Data pruning affects *which* samples are seen, and early stopping uniformly reduces the number of times *all* samples are seen. Does this qualitative difference bear out quantitatively?

We study this by equalizing the number of samples seen by the data pruning methods against the full baseline trained for 200 epochs. This is achieved by running the pruning methods for $N/k$ epochs while adjusting the learning rate scheduler accordingly. Fig. 5 shows the results of these experiments. First, note that the full baseline acts as an experimental upper-bound over all the other methods. This corroborates a key finding of our work. Contrary to what is suggested by the prior work on static pruning, there is not a subset of the data that is most informant about the true decision boundary. If such a subset were to exist, then allocating the full samples seen budget on that subset would converge faster than the full baseline. As Fig. 5 shows, the static baselines that follow this allocation under-perform, suggesting that the always set is necessary but not sufficient for full performance. On the other hand, the dynamic methods track the full baseline more closely by training on a mix of the always and sometimes sets. Notably, we can see the difference between the three methods proposed in the paper. While all methods achieve the same final accuracy, the UCB and $\epsilon$-greedy methods out-perform the uncertainty method with fewer samples seen. By the end of training, all methods have a stable enough estimate of the score distribution to converge to the full accuracy. Furthermore, this analysis shows the qualitative difference between data pruning and early stopping bears out. For the same number of samples seen, the full baseline can train on every sample in the dataset but for fewer iterations. The model sees a super-set of the samples seen by any data pruning method, and as a result, it acts as an upper-bound on the convergence.

## 4.4 CIFAR-100 results

Fig. 6 shows the result of dynamic data pruning on CIFAR-100. We retained the same hyper-parameters as in the CIFAR-10 result but omit the uncertainty method without the EMA as its performance was unsatisfactory. We can see that EL2N and forget scores are able to nearly retain the full baseline performance up to a pruning rate of 20% but deteriorate rapidly beyond that point.

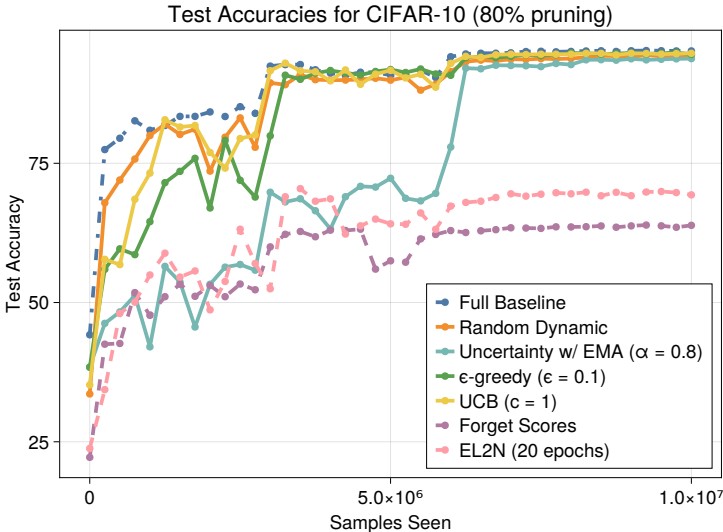

Figure 5: Dynamic data pruning CIFAR-10 with ResNet-18 at 80% pruning, showing the test accuracy as a function of total samples seen during training for each method. Static methods clearly fail to reach the full accuracy, even when equalizing for samples seen against the full baseline. All dynamic methods reach the same accuracy as the baseline, but at different rates—highlighting how methods like UCB can more accurately estimate the score distribution with fewer observations.

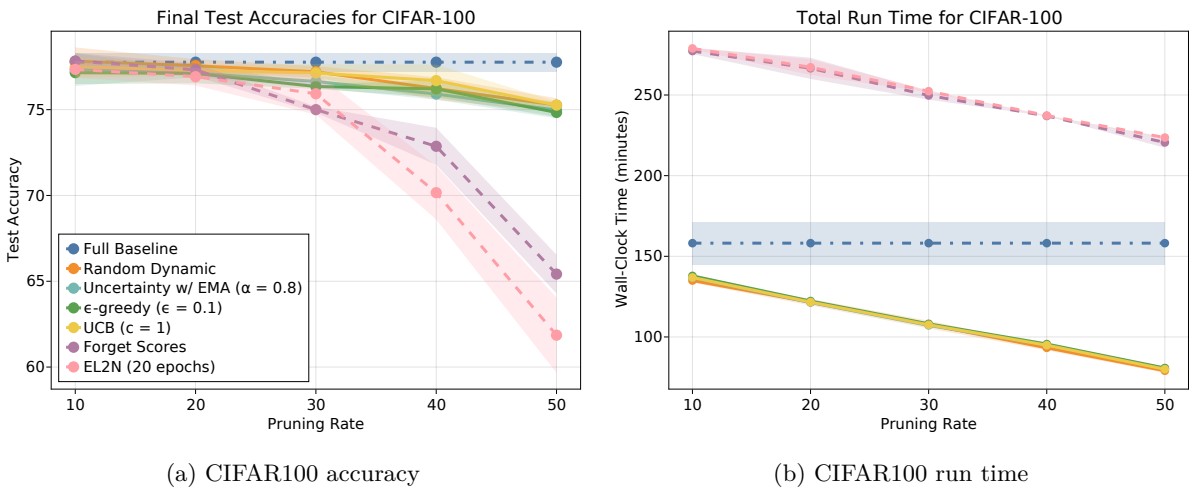

(a) CIFAR100 accuracy

(b) CIFAR100 run time

Figure 6: Dynamic data pruning CIFAR-100 with ResNet-34. Fig. 6a shows the final test accuracies for each method and Fig. 6b shows the run time to execute the approach (including scoring cost). Zoomed version available in supplementary material.

All the dynamic methods outperform the static baselines. While the UCB does appear to have slightly higher accuracy than the random dynamic method at more aggressive pruning rates, most of our methods underperform compared to the random dynamic method. This is not surprising when we consider the sample selection distribution in Fig. 7.

Unlike CIFAR-10, CIFAR-100 has almost no never points and is far more uniform in its sample selection distribution. This indicates that there is very little opportunity to select samples more intelligently than a random dynamic method. One plausible explanation for the lack of never samples is the low number of

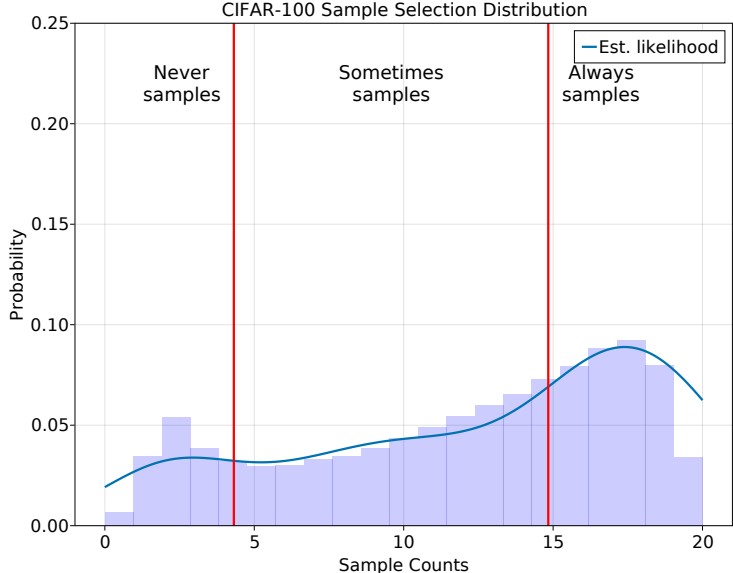

Figure 7: Sample selection distribution on CIFAR-100 at 40% pruning with UCB. Unlike CIFAR-10, CIFAR-100 has very few never samples, which means that the always and sometimes samples takes a large portion of the dataset. Thus, a random dynamic method can prune close to optimally on this dataset.

samples per class in CIFAR-100 (500 vs. 5000 for CIFAR-10). Thus, each sample has more influence on the decision boundary, and the intrinsic redundancy in the dataset is much lower.

### 4.5 Imbalanced CIFAR-10 results

So far, we have studied standard datasets which are well balanced. Our CIFAR-100 results suggest that the number of samples per class can have a large influence on the pruning opportunity. To further explore this phenomenon, we generate a synthetically imbalanced variant of CIFAR-10 by sub-sampling each class according to Tab. 2.

Table 2: The subsampling rates by class for the imbalanced CIFAR-10 dataset.

| Class Indices | Subsample Rate |
| --- | --- |
| 0, 1 | 25% |
| 2, 3, 4 | 50% |
| 5, 6 | 75% |
| 7, 8, 9 | 100% |

The results are shown in Fig. 8a. Our methods outperform the random dynamic method at aggressive pruning rates, but all methods lose performance at modest pruning rates. This is expected based on Fig. 8b. The lack of always/never samples makes this dataset difficult to prune relative to a dataset like CIFAR-10. Unlike CIFAR-100, there does seem to be a larger portion of never samples and consequently more opportunity for pruning. The UCB method is able to target this opportunity once the pruning rate is $> 50\%$.

### 4.6 Downsampled CIFAR-10 results

Similar to the imbalanced CIFAR-10 dataset, we also study a downsampled variant of CIFAR-10. We decrease the number of samples per class from 5000 to 1000 for every class. This mimics CIFAR-100 where the number of samples per class is 500.

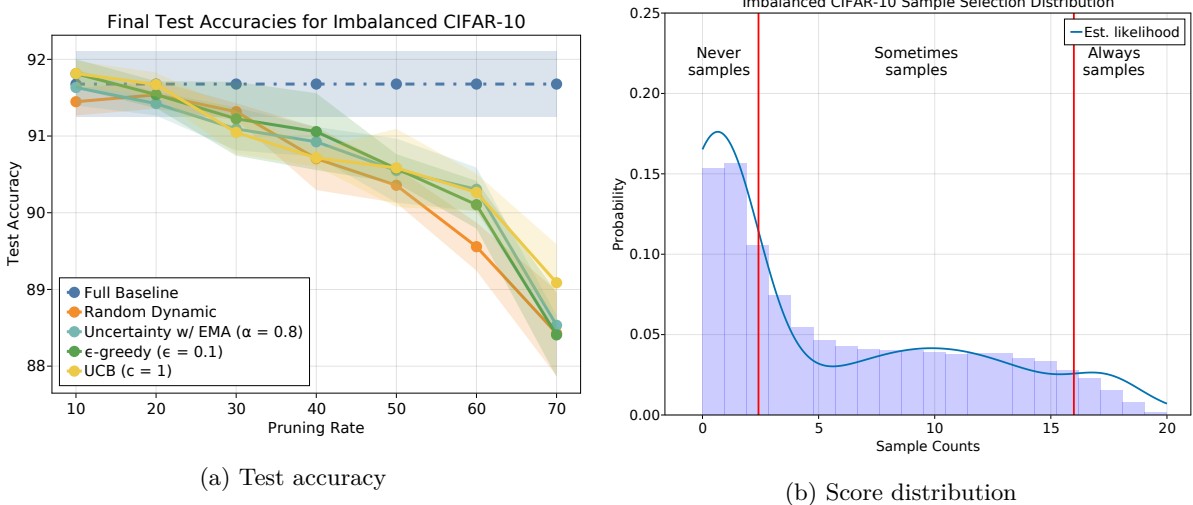

(a) Test accuracy

(b) Score distribution

Figure 8: (a) Dynamic data pruning a synthetically imbalanced CIFAR-10 with ResNet-18. On imbalanced data, our proposed methods, like UCB, are able to obtain higher performance than a random dynamic method. (b) Sample selection distribution on the imbalanced CIFAR-10 dataset at 40% pruning with UCB. The lack of always samples contributes to lower accuracies across all methods, but UCB is still able to obtain a slight competitive edge by delineating samples into the three regions more accurately.

The results are shown in Fig. 9a. Similar to the CIFAR-100 results, the uncertainty with EMA and $\epsilon$-greedy methods are worse than the random dynamic method. Only the UCB method matches the random dynamic method in performance. These results are corroborated by Fig. 9b which shows that the sample selection distribution follows the same trend as CIFAR-100.

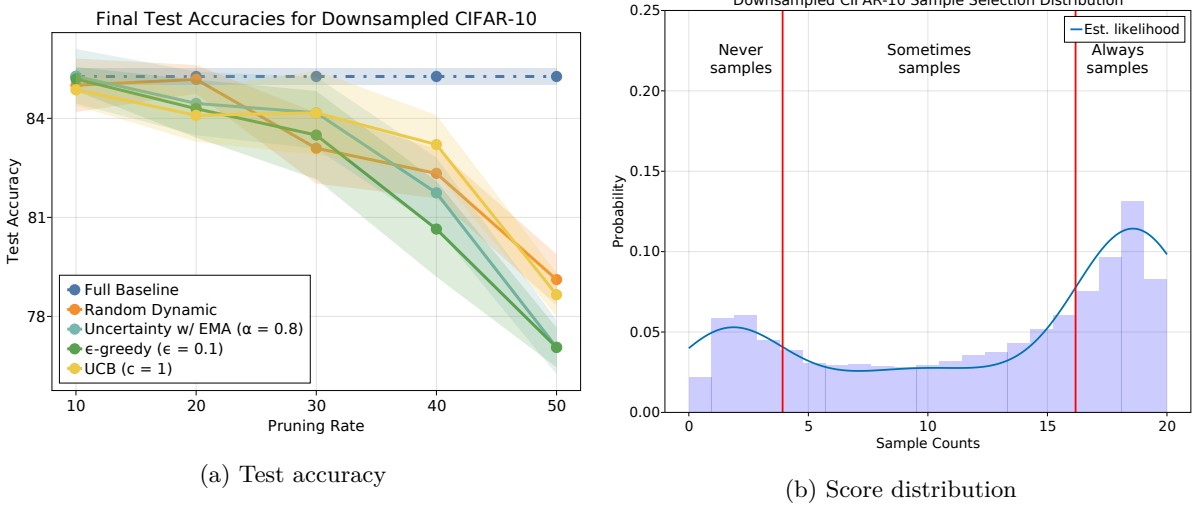

(a) Test accuracy

(b) Score distribution

Figure 9: (a) Dynamic data pruning a synthetically downsampled CIFAR-10 with ResNet-18. This dataset mimics CIFAR-100 where each sample is more important due to the low samples-per-class. Like CIFAR-100, the random dynamic method performs just as well or better as the other methods, since the nature of the dataset presents little opportunity for pruning. (b) Sample selection distribution of samples on the downsampled CIFAR-10 dataset at 40% pruning with UCB. The density function follows a similar shape to CIFAR100 where there are very few never samples.

### 4.7 CINIC10

So far, the datasets, CIFAR10 and CIFAR100, we have looked at have been fairly smaller scale. To test our methods on a larger dataset, we use CINIC10 as a testbed (Darlow et al., 2018). CIFAR10 and CIFAR100 are on opposite ends of difficulty: CIFAR10 has 6000 examples per class whereas CIFAR100 has 600 examples; CINIC10 is in between both of these datasets in terms of difficulty. It has 270,000 images (4.5 ×) and we combine the training and validation set (still maintaining a separate test set). It has images sourced from both ImageNet as well as CIFAR10. We apply the same methodology and training framework as we used for all the other datasets.

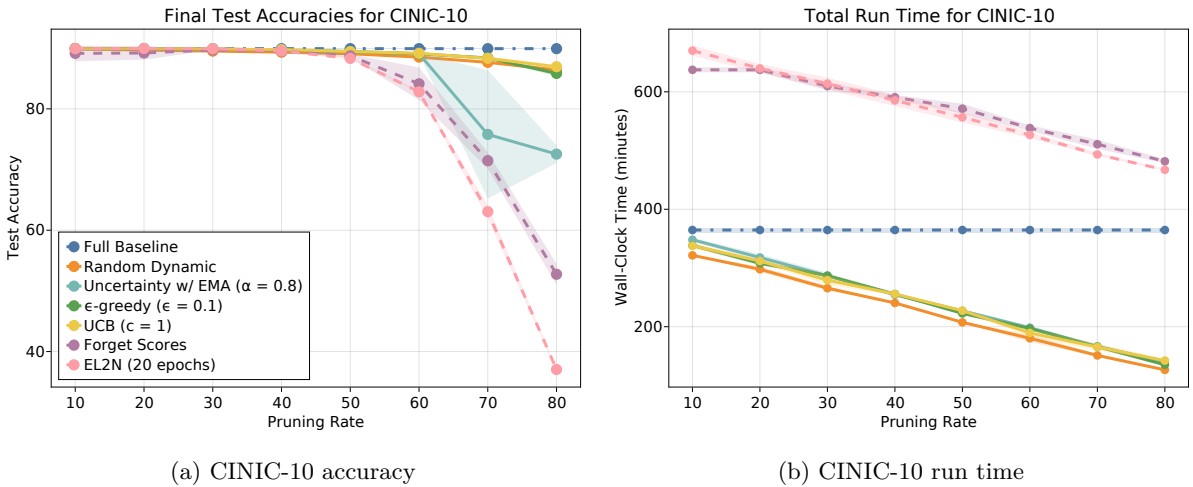

(a) CINIC-10 accuracy    (b) CINIC-10 run time

Figure 10: Dynamic data pruning applied to CINIC-10 with ResNet-18. Fig. 10a shows the final test accuracies for each method and Fig. 10b shows the run time to execute the approach (including scoring cost). Zoomed version available in supplementary material.

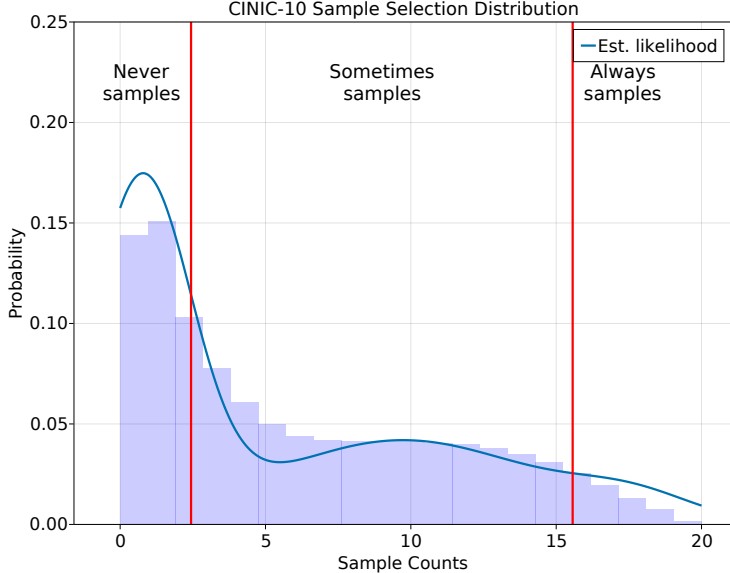

Figure 11: Sample selection distribution on the CINIC10 dataset at 70% pruning with UCB. Similar to the imbalanced CIFAR10 dataset, UCB is able to target the never samples and cycle through the sometimes samples to maintain high accuracy even at an aggressive pruning rate.

As shown in Fig. 10, our methods reduce the training cost by nearly 5× at 80% pruning while remaining close to the baseline accuracy. Similar to all other datasets, the UCB method is superior compared to the other dynamic pruning strategies. In comparison, the static methods begin to drop down in accuracy at a pruning rate of 60% and then rapidly drop off as the pruning rate is increased. Fig. 11 shows the sample selection distribution of UCB at a pruning rate at 70% which follows the trends we see in other datasets—we can correctly select the sample by looking at the variance in the scores. The main take-away is that our methods are able to outperform static approaches by effectively targeting the sometimes samples during training.

### 4.8 ImageNet

We trained ResNet18 on the ImageNet dataset on a NVIDIA GTX A100 GPU. We use the ffcv data loader to compress the dataset and reduce the time it takes to stream the images to the model. We use a basic step learning rate scheduler, set the batch size to 1024, use standard data augmentation and train for 90 epochs. Other configuration details are the same as the previous experiments. We note that this experiment builds upon the prior work by evaluating the static methods on ImageNet.

We see in the Fig. 12 that among the static methods that EL2N performs the worst dropping at 40% pruning and then further at 60%. Forget scores are able to do surprisingly well, tracking with all the dynamic methods until 50% pruning. At 70% pruning, the forget score method drops several percentage points below the dynamic methods which may not be acceptable for ImageNet. Notably, these results show that even after ignoring the cost of scoring, the static methods can only provide a 1.5-2x speedup on ImageNet, further underscoring the role of sometimes samples in any future data pruning work.

The random dynamic method outperforms all other dynamic methods. This could be due to ImageNet samples having multiple possible labels which will affect our methods' ability to find characteristic samples for a dataset. When we plot the top-5 accuracies for all methods, we see that our dynamic methods are able to do better than random with the exception of UCB at 70%.

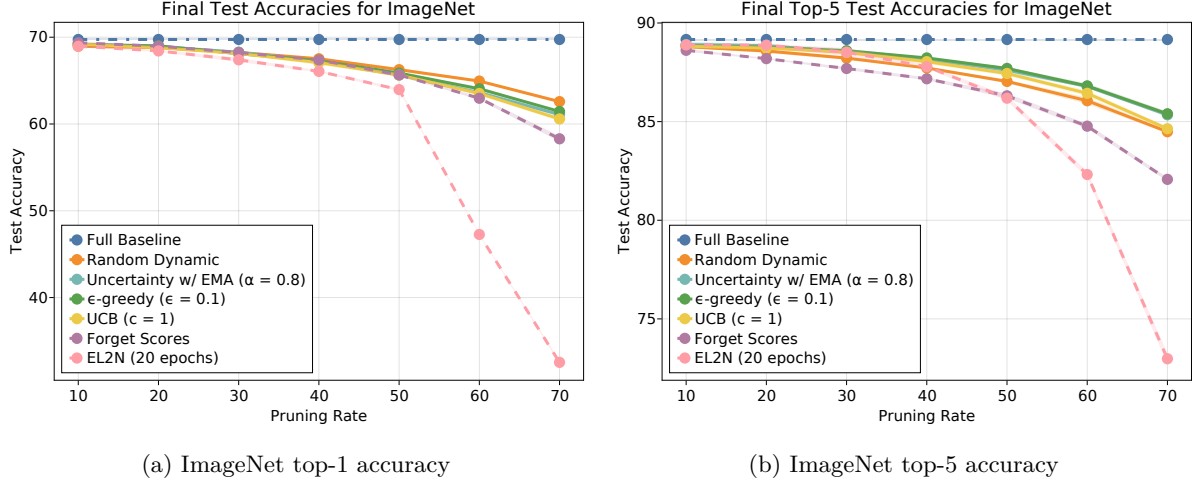

(a) ImageNet top-1 accuracy        (b) ImageNet top-5 accuracy

Figure 12: Dynamic data pruning ImageNet with ResNet-18 with 90 epochs, showing the final test accuracy for each method. Our methods outperform all the static baselines, but the forget score baseline does better than expected.

### 4.9 Influence of hyperparameters

Our methods have several hyper-parameters that can be adjusted—the pruning rate ($k/N$), the pruning period ($T_p$), the exploration rate ($\epsilon$; $\epsilon$-greedy only), and the confidence ($c$; UCB only). Our previous results fix most of these hyper-parameters and vary only the pruning rate. We also conducted experiments where we sweep each hyper-parameter. The overall conclusion is that the remaining hyper-parameters have minimal effect on the end result. The hyper-parameter sweeps are reported in the supplementary material.

# 5 Conclusion

This work presents a dynamic approach to data pruning. We reframe the problem as a decision making process, and we present three methods (uncertainty with EMA, $\epsilon$-greedy, UCB) for pruning datasets online while training. This allows our methods to be applied to novel datasets that are being trained on for the first time. Unlike the prior work, our approaches do not incur significant overhead, and we are able to substantially reduce the total run time.

Moreover, we introduce the notion of a sample selection distribution which separates a dataset into three groups—always samples, sometimes samples, and never samples. We show that sometimes samples are difficult to rank, and as a result, static pruning methods cannot effectively select a subset of them. Surprisingly, we find that some datasets, like CIFAR-100, have little opportunity for sophisticated pruning. This is because the dataset has very few always/never samples. In these cases, the best option is to prune the dataset using a random dynamic method, since all samples have equal importance. We corroborate these conclusions by testing these hypotheses on synthetically modified CIFAR-10 datasets with severe class imbalance or low samples-per-class.

We hope our work emphasizes the need to understand data pruning as a function of the dataset, model, and training trajectory. By viewing the pruning problem as an online decision-making process, we expect future work to borrow from active learning and reinforcement learning to more effectively target sometimes samples. In lieu of these improvements, our methods bring practical, efficient data pruning to DL researchers.

## 5.1 Societal impacts

This work has financial and environmental societal implications. Current trends in deep learning have made training state-of-the-art models prohibitively expensive. As a result, only large research organizations have access to large, overparameterized models. By reducing the total training time, our methods allow independent researchers to train more sophisticated models. Moreover, unlike the prior work, our methods can be applied directly to novel datasets.

More importantly, large-scale deep learning requires weeks of compute time on an energy-hungry GPU cluster. Many of these GPUs are used to parallelize iterating over the dataset. By pruning the data, we allow models to be trained with fewer GPUs or in less time. Both outcomes translate to lower energy consumption and carbon footprints for DL training.

On the other hand, by helping to democratize complex DL models to new domains, our work may negatively impact areas of society affected by those domains. Namely, since bias and robustness of DL is poorly understood, an increase in its applicability may cause unexpected harm.

## 5.2 Limitations and future work

Our extensive experiments show that dynamic data pruning is an effective approach for accelerating deep learning training; however, there are limitations to our techniques. As the proposed methods rely upon the per-sample loss, label noise would hurt the accuracy of the classifier significantly more than the full baseline as each individual sample has more influence on the decision boundary.

Additionally, our proposed methods fail to outperform a random dynamic baseline for datasets with many sometimes samples, like CIFAR-100. In these contexts, a more thorough understanding of sometimes samples could lead to a more sophisticated approach that improves over random. In particular, we separate the dataset into the three regions qualitatively. A more principled, formal approach to labeling each samples as "always," "sometimes," or "never" could help tailor pruning methods to the specific dataset.

Moreover, our methods only make a decision about which samples to keep at each checkpoint, but future methods could include various hyperparameters as part of the decision making process. For example, an algorithm could decide the samples to keep at the current checkpoint, as well as the period until the next checkpoint or the pruning rate at the current checkpoint. Such methods could iteratively prune the dataset leading to a better final test accuracy.

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

## A  Run time breakdown for static methods

Fig. 13 and Fig. 14 shows the wall-clock time of the static methods with and without the pre-training scoring cost. The static methods achieve roughly a similar run time to the random dynamic method, which is to be expected since there is no overhead at each checkpoint.

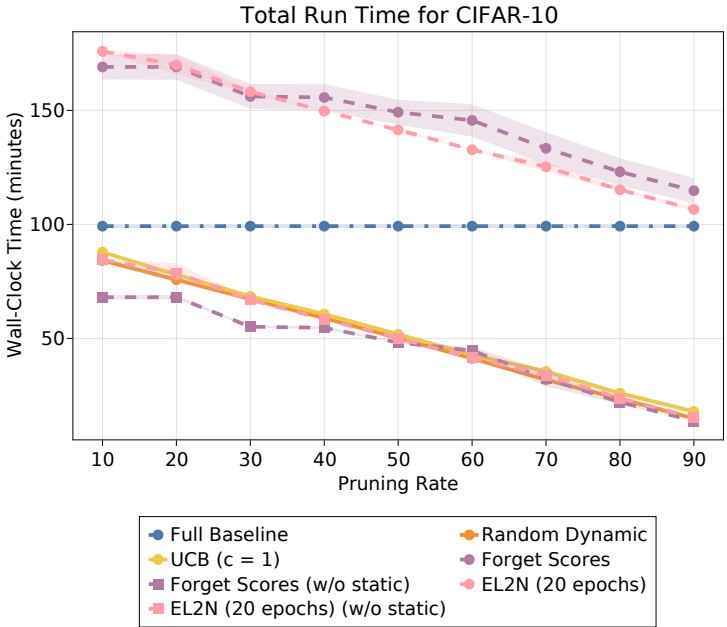

Figure 13: Dynamic data pruning on CIFAR-10 with pre-training scoring cost included and excluded for static methods.

## B  Static approaches

In our dynamic methods, we maintain running means and variances of the scores. In contrast, the static EL2N method takes an offline mean across many model initializations. A natural question is whether our methods can be applied in a static fashion for the same accuracy gains. The results in Fig. 15 show that this is not the case.

When we statically apply the UCB algorithm to various model initialization trials in the EL2N method, we see no improvement in performance. This makes sense, because the variance in the UCB method is used to dynamically sample under-observed points. In a static variation, this feature of the algorithm is unused.

If we apply an $\epsilon$-greedy approach to the static EL2N policy (i.e. every checkpoint we select $1 - \epsilon$ fraction of the samples using the pre-computed EL2N scores, and we select $\epsilon$ fraction randomly), we see an increase in accuracy. Therefore, we see that even a small amount of dynamism improves static scores. The boost in accuracy still under-performs compared to the random dynamic method.

## C  Zoomed version of Dynamic Pruning Plots

In this section, we plot zoomed versions of the the pruning plots in the main body to see the differences between the dynamic methods. Fig. 16 shows the difference between all the dynamic methods, with the static methods omitted; all the active/reinforcement learning methods are able to outperform the random dynamic method. Fig. 17 is the same experiment but with 100 fewer epochs and shows similar conclusions. Fig. 18 shows a zoomed in version of data pruning on CIFAR-100 with no static methods, showing that the

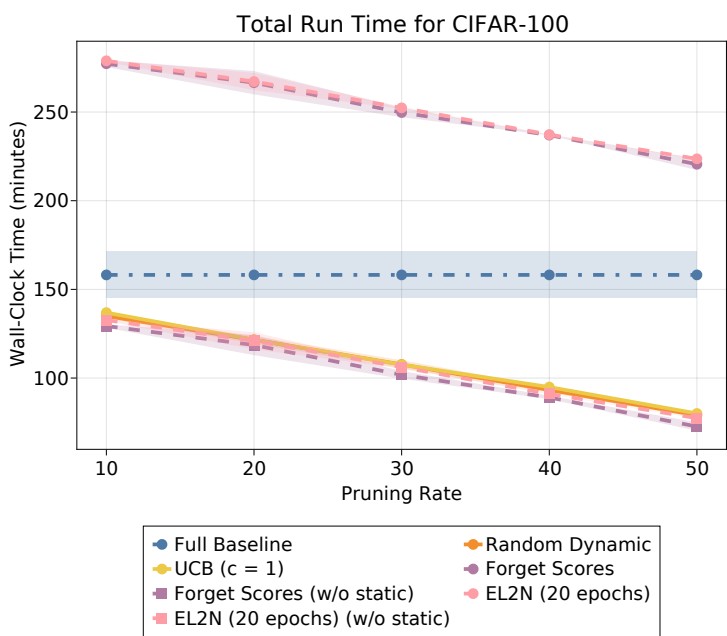

Figure 14: Dynamic data pruning on CIFAR-100 with pre-training scoring cost included and excluded for static methods.

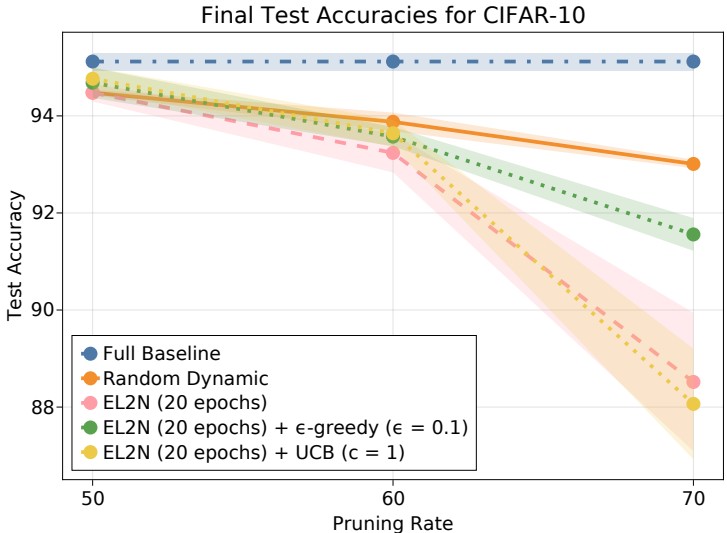

Figure 15: Apply our RL methods on top the static EL2N scores. "EL2N + $\epsilon$-greedy" uses the static EL2N scores at each checkpoint but also selects $\epsilon$ fraction of the samples randomly. "EL2N + UCB" applies the UCB algorithm statically to the EL2N scores from each model initialization trial.

random dynamic method outperforms most of the other methods. Fig. 19 shows a slice of the pruning rates between the dynamic methods and demonstrates that UCB outperforms it at every pruning rate.

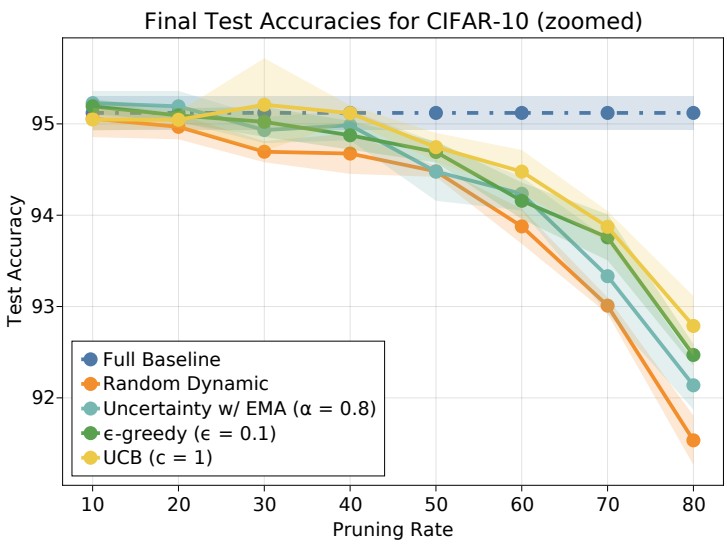

Figure 16: Zoomed in plot of dynamic data pruning on CIFAR-10 with ResNet18 with 200 epochs.

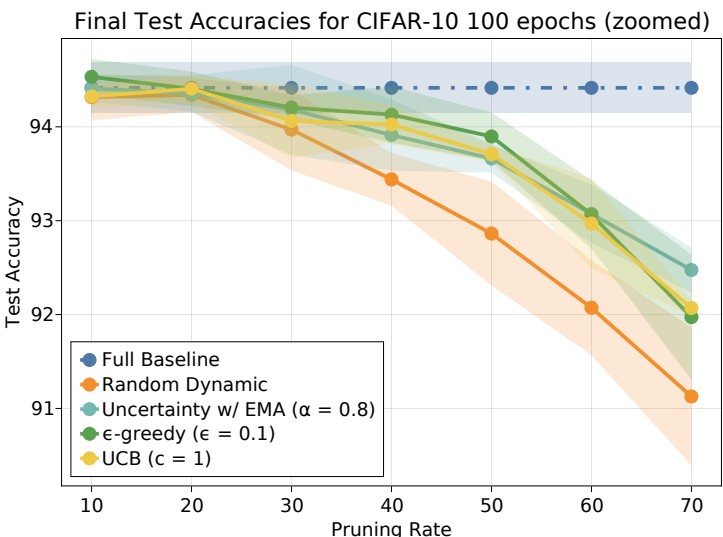

Figure 17: Zoomed in plot of dynamic data pruning on CIFAR-10 with ResNet18 with 100 epochs.

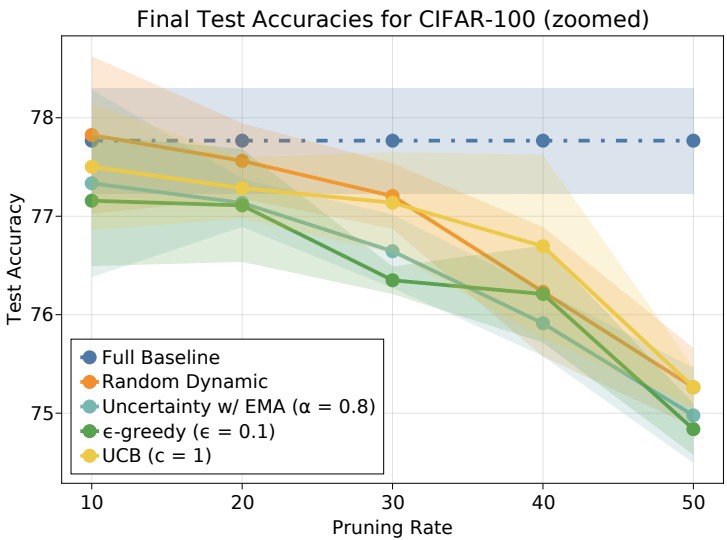

Figure 18: Zoomed in plot of dynamic data pruning on CIFAR-100 with ResNet34 with 200 epochs.

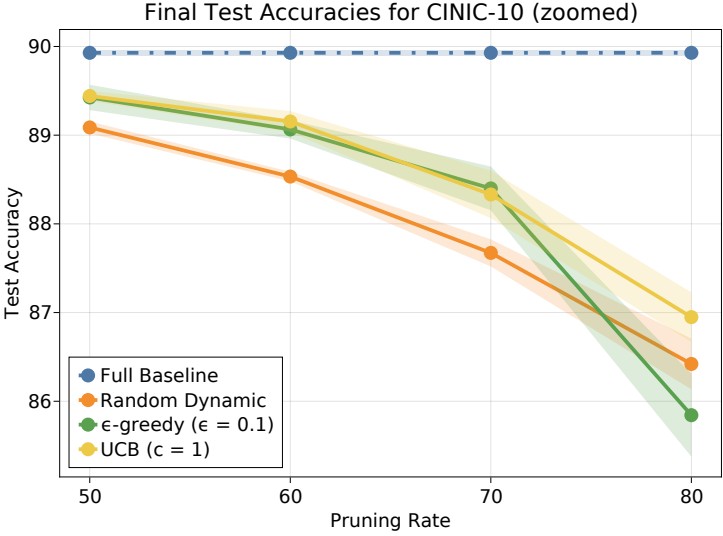

Figure 19: Zoomed in plot of dynamic data pruning on CINIC-10 with ResNet18 with 200 epochs.

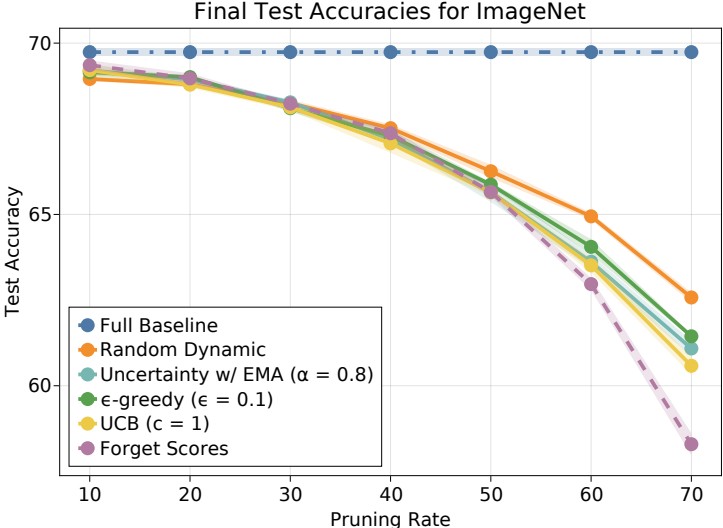

Figure 20: Zoomed in plot of dynamic data pruning on ImageNet with ResNet18 with 90 epochs.

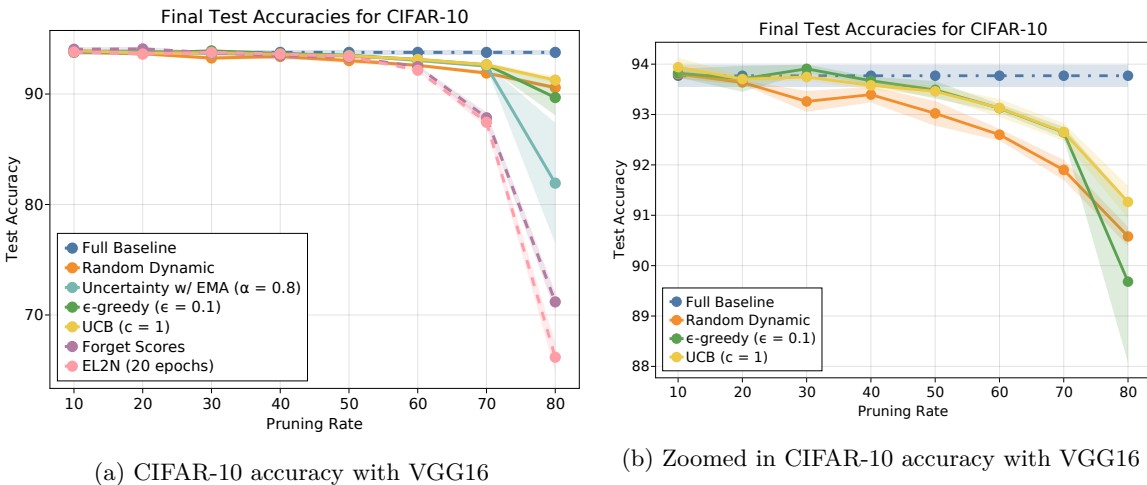

(a) CIFAR-10 accuracy with VGG16

(b) Zoomed in CIFAR-10 accuracy with VGG16

Figure 21: Re-evaluating our CIFAR-10 experiments with VGG16 instead of ResNet18. We see that the model architecture does affect the accuracy-pruning curves significantly.

## D   Additional model architectures

To test out our methods performing on different models, we repeat our experiments with VGG16 on CIFAR10. Fig. 21 shows our methods are able to outperform the static pruning methods at higher pruning rates and the UCB method still maintains the highest performance over all the other dynamic methods. Uncertainty with EMA and $\epsilon$-greedy are able to keep pace with UCB until 80% pruning, where the performance drops below random dynamic. When we compare the margin between different methods on different architectures, we can see that it has a significant impact on how much pruning can be done.

## E   Transferring datasets pruned with smaller networks

The authors of the forget score work mention (but do not test) whether a simpler network can be used to prune the dataset. In order to test whether we can use a smaller architecture, we train a model with 2

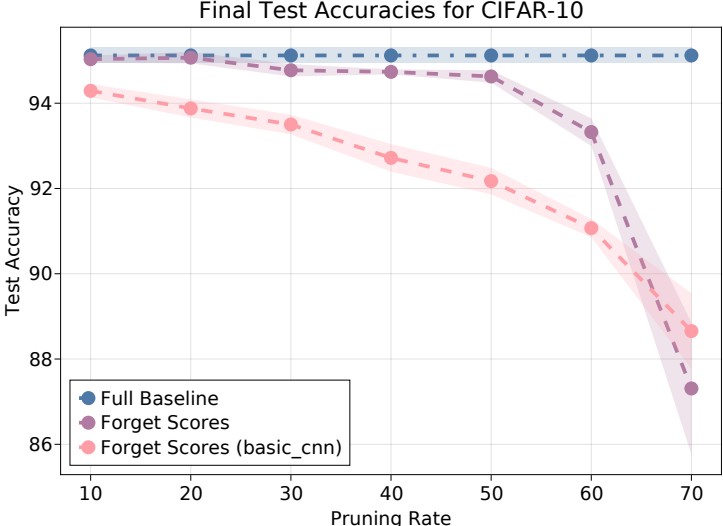

Figure 22: Statically pruning CIFAR-10 using a smaller network results in worse performance. This emphasizes the need for dynamic methods on novel datasets.

convolutional layers and 2 full-connected layers, called `basic_cnn`, with the same experimental settings and track the forget scores during training. The training overhead was only 48 min compared to the 92 min overhead with ResNet18. We then retrain the model with subsets and plot it compared to the ResNet18 forget scores. Based on Fig. 22, it is clear that using a smaller architecture yields worse approximations for forgettable samples. Therefore, it seems like one must use a model that is similar in size and architecture in the pre-training phase to maintain high accuracy.

## F    Hyperparameter search

Fig. 23 shows the results of sweeping the pruning period ($T_p$), the exploration rate ($\epsilon$; $\epsilon$-greedy only) and confidence ($c$; UCB only). The pruning period of 10 epochs provides the best performance vs. wall-clock time tradeoff for all methods. Varying the exploration rate does not impact the accuracy, suggesting that only a small amount of randomness is required. Varying the confidence suggests that incorporating variance into the scoring mechanism is sufficient to increasing the performance, agnostic of $c$'s value.

## G    Example always, sometimes, and never samples

In order to see the qualitative difference between always, sometimes, and never samples, we take a random selection of each set from CIFAR-10 and plot them in Fig. 24. There appears to be no discernible difference in the various subsets.

## H    Reproducing the results

To run the code, change directories to the `dynamic_data_pruning_code` folder. Follow the instructions in the `README.md` in the `src` folder to re-create the training environment and run each experiment.

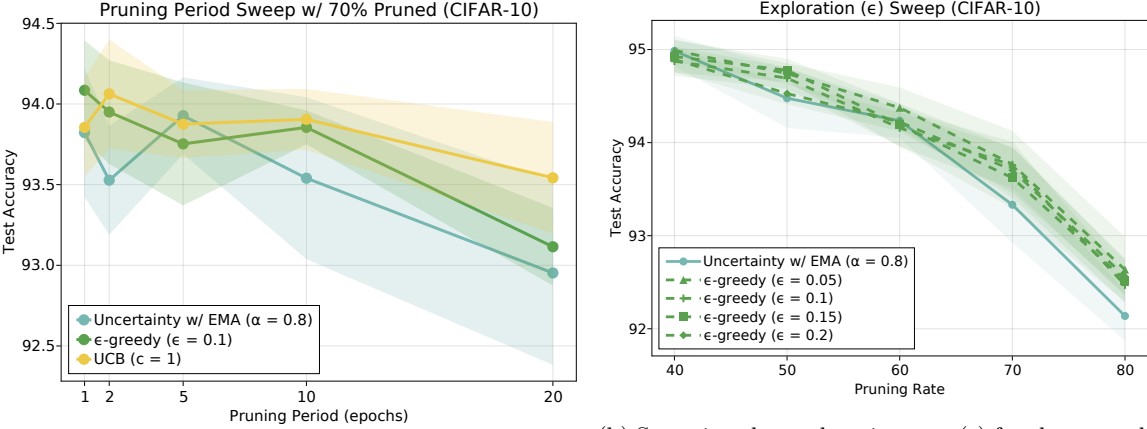

(a) Sweeping the frequency for all methods.

(b) Sweeping the exploration rate ($\epsilon$) for the $\epsilon$-greedy method.

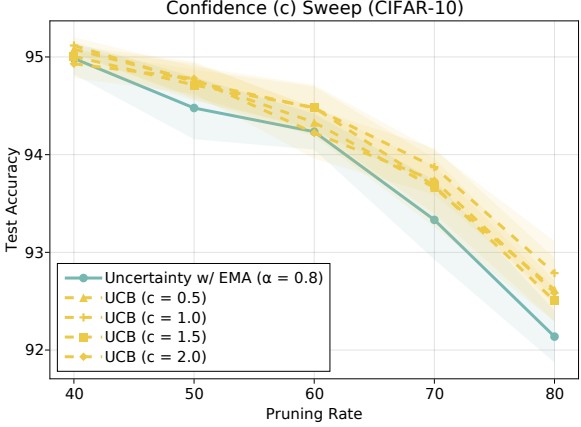

(c) Sweeping the confidence ($c$) for the UCB method.

Figure 23: The effect of sweeping various hyper-parameters while training on CIFAR-10.

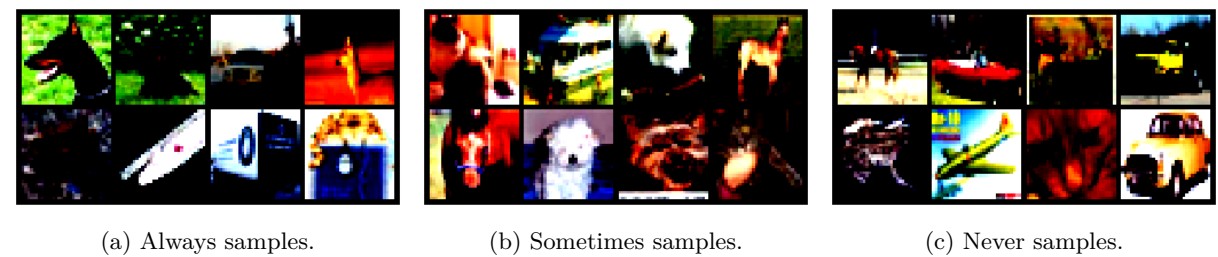

(a) Always samples.      (b) Sometimes samples.      (c) Never samples.

Figure 24: Examples of always, sometimes, and never samples taken from CIFAR-10. We do not notice a qualitative difference between the three groups.

