# OpenReview forum: "Accelerating Deep Learning with Dynamic Data Pruning"
_TMLR — Rejected by TMLR_

### Review · Reviewer_MUb4 · 2022-08-28

**Summary Of Contributions:**

Summary.

This paper is dedicated to designing dataset pruning algorithms that aim to trim down the number of training iterations by only using part of the training datasets. The authors propose a dynamic data pruning method that borrows the classic design philosophy of network sparse training. They find the uniform random dynamic pruning reaches superior performance at an aggressive pruning rate. The authors explain that the performance improvements come from the adaptive importance of each sample during the full training procedure.

**Broader Impact Concerns:**

The broader impact concerns are well discussed.

**Requested Changes:**

Refer to the weakness section.

**Strengths And Weaknesses:**

Pros.

1. The authors ablation diverse picking criteria, as shown in Figure 2. It provides sufficient comparisons.

2. The paper is well-written and easy to follow.

3. The dynamic data pruning is quite interesting.



Cons.

1. Visualizations of the always / sometimes / never samples will be very helpful. I am very curious about the visual contents in the never samples.

2. Why do the authors provide three different pruning selection criteria that perform similarly? As shown in Figures 2 and 4, they achieve almost the same performance, considering the error bar. Thus, I think the proposals are redundant.

3. The experiments are not sufficient, since they are conducted on limited datasets and network backbones. More large-scale datasets and advanced/diverse networks are needed. For example, ImageNet for the datasets and MobleNet/VGG for the network backbones.

4. What is the overlap of selected subsets between different pruning selection criteria?

5. Can the selected subsets transfer across different network architectures? This point will make the data pruning more meaningful.

6. The related works are not sufficient, especially for active learning (only one reference) and coreset. More discussions about existing works in these fields are required. Also, the references are quite old, where only 3-4 papers from 2021 and 2022.

---

> ### Author Response · Authors · 2022-09-22
> **Initial response**
>
> Thank you for your review. Below are some specific comments to the points that you have raised. We are working to generate figures addressing the remaining concerns as well.
>
> > Why do the authors provide three different pruning selection criteria that perform similarly? As shown in Figures 2 and 4, they achieve almost the same performance, considering the error bar. Thus, I think the proposals are redundant.
>
> The difference between the strategies is dependent on the dataset being pruned. This can be observed in the zoomed-in plots in the supplementary material. We suggest focusing on the rightmost data points where the pruning rate is highest. For CIFAR-10 (Fig. 16-17, revised), each method introduced provides an additional gain in accuracy with UCB performing the best. For CIFAR-100 (Fig. 18, revised), where the dataset is dominated by sometimes samples, all methods perform as well as random. So, while the proposals are redundant for datasets with many sometimes samples, they can provide gains at extreme pruning rates on datasets with enough always and never samples.
>
> Additional results as suggested by the other reviewers further reveals the difference between the methods. In Fig. 5 (revised), we see that while all the methods have the same final accuracy, the UCB method converges the fastest w.r.t. samples seen. So, if we want to combine data pruning with a short number of epochs, it would be better to use UCB over the uncertainty w/ EMA method.
>
> > The experiments are not sufficient, since they are conducted on limited datasets and network backbones. More large-scale datasets and advanced/diverse networks are needed. For example, ImageNet for the datasets and MobleNet/VGG for the network backbones.
>
> Below are some additional results that we can add to the paper to address this concern. For ImageNet, none of the prior work is evaluated on this dataset which is why we omitted it originally. Our results below also evaluate the prior work for the first time on ImageNet. These results are included in the revised paper Fig. 12 and 21.
>
> We see in Fig. 12 that EL2N performs the worst dropping at 40% and then further at 60%. Forget scores are able to do surprisingly well, tracking with all the dynamic methods until 50% pruning. Notably, these results show that even after ignoring the cost of scoring, the static methods can only provide a 1.5-2x speedup on ImageNet, further underscoring the role of sometimes samples in any future data pruning work. All the other dynamic methods underperform the random dynamic method which maintains the highest performance from 40% pruning on. This could be due to ImageNet samples having multiple possible labels which will affect our methods’ ability to find characteristic samples for a dataset. When we plot the top-5 accuracies for all methods, we see that our dynamic methods are able to do better than random with the exception of UCB at 70%.
>
> To test out our methods performing on different models, we repeat our experiments with VGG16 on CIFAR10. Fig. 22 shows our methods are able to outperform the static pruning methods at higher pruning rates and the UCB method still maintains the highest performance over all the other dynamic methods. Uncertainty with EMA and eps-greedy are able to keep pace with UCB until 80% pruning, where the performance drops below random dynamic. When we compare the margin between different methods on different architectures, we can see that it has a significant impact on how much pruning can be done.

---

> > ### Author Response · Authors · 2022-09-29
> > **Continued response**
> >
> > > Visualizations of the always / sometimes / never samples will be very helpful. I am very curious about the visual contents in the never samples.
> >
> > This has been added in the revised supplementary section (Fig. 24). We do not see any qualitative difference between the various groups.
> >
> > > What is the overlap of selected subsets between different pruning selection criteria?
> >
> > We compared the average ranking of the samples for CIFAR-10 across all trials by taking pairwise Spearman correlations. The correlation coefficient was $\approx 10^{-2}$ suggesting weak overlap. A more informative comparison of the methods is to compare their converge over the course of training, as shown in Fig. 5 (revised). These results highlight how the UCB and $\epsilon$-greedy method improve the convergence vs. the uncertainty w/ EMA.
> >
> > > Can the selected subsets transfer across different network architectures? This point will make the data pruning more meaningful.
> >
> > Transferring across architectures is possible, but this will not attain the same accuracy as the original. This is because only the always samples can be statically transferred. A key finding of our work is the non-negligible contribution of sometimes samples which must be dynamically selected (i.e. non-transferrable). Transferring only the always samples and continuing to dynamically select the sometimes samples does achieve close to the original accuracy. We show this in Sec. 4.2 of the revised paper.
> >
> > > The related works are not sufficient, especially for active learning (only one reference) and coreset. More discussions about existing works in these fields are required. Also, the references are quite old, where only 3-4 papers from 2021 and 2022.
> >
> > We have updated the related work section with more references and explanation.

---

### Review · Reviewer_2RVp · 2022-08-28

**Summary Of Contributions:**

This article accelerates training by selecting proper subset of the dataset on runtime. The main method is to insert checkpoints between epochs, re-evaluating the importance of each sample and adjusting the subset used by the next check period. Contributions are 1) Compared to prior static data pruning, the author uses a runtime method to select key samples. It’s good for users to train models in a new dataset. 2) Analyze the difference about always samples, sometimes samples and never samples, providing a possible reason for better accuracy than that of static pruning.



**Requested Changes:**

NA

**Strengths And Weaknesses:**

pros:
1. The insight about sometimes samples is novel and attractive.
2. The paper has detailed experiments and promising test results.
cons:
1. The acceleration results compared with static data pruning is not persuasive enough since articles such as forgetting scores said a simpler net could be used to prune the dataset before training large models.

---

> ### Author Response · Authors · 2022-09-22
> **Initial response**
>
> Thank you for your review. Below are some specific comments to the points that you have raised.
>
> > The acceleration results compared with static data pruning is not persuasive enough since articles such as forgetting scores said a simpler net could be used to prune the dataset before training large models.
>
> The authors of the forget score work mention (but do not test) whether a simpler network can be used to prune the dataset. In order to test whether we can use a smaller architecture, we train a model with 2 convolutional layers and 2 full-connected layers, called basic_cnn, with the same experimental settings and track the forget scores during training. The training overhead was only 48 min compared to the 92 min overhead with ResNet18. We then retrain the model with subsets and plot it compared to the ResNet18 forget scores. Based on Fig. 22 (added to revised paper), it is clear that using a smaller architecture yields worse approximations for forgettable samples. Therefore, it seems like one must use a model that is similar in size and architecture in the pre-training phase to maintain high accuracy.

---

### Review · Reviewer_Nr2k · 2022-09-16

**Summary Of Contributions:**


This paper proposes to dynamically subsample training sets for supervised DNNs every few epochs based on some function of the cross entropy loss and a pruning ratio.

Three methods are proposed:
- Keep a moving average of the cross entropy for each point to sort examples and use the ones with the highest loss
- As before but take points from the remaining data with probability $\epsilon$
- Also keep a moving average of the variance of the CE and sort according to mean + constant * variance (UCB style).

Experiments with these methods on CIFAR10 and CIFAR100 suggest that when using large pruning ratios, it is better to use such dynamic/per-epoch pruning strategies rather than strategies suggested by prior work which decide which data to keep only once, before training the model.

Since training on less data (due to the pruning) takes less time, the authors therefore suggest that for resource constrained training, the proposed approaches should be desirable.

A central claim of the paper is that while prior works filter datasets beforehand, it is always advantagous for the filtering to depend on the current state of the model.

**Broader Impact Concerns:**

The broader impact discussion is good, and highlights the issues created by the imbalance of access to compute power in the research community.

**Requested Changes:**

In order of importance, I recommend the following changes
- Improving experiments to make sure that the observed results are consistent, and are not caused by simply seeing less data.
- Clarify or remove the use of RL as a justification for the proposed methods.
- Clarify the algorithm's use of the proposed scores.
- Discuss additional related literature, ideally comparing to it experimentally.

#### Comments on text & figures

> per-iteration penalty of evaluating

I get the intended meaning, but "evaluating" can also mean running the validation. I'd just say "the per-iteration computational cost of the model".

Fig 1.; it wasn't immediately clear the the $x$ indices changed from row to row, I'd recommend in addition using a unique shape (circle, square, oval, star, etc.) for each point, this way the difference in each row pops out.

> By counting every time each sample is selected across all scoring checkpoints, we find that a dataset can be qualitatively split into three groups

Selected by which method? The proposed method or prior work? If the proposed method by design does this, then this statement feels tautological. After a full read, it should be made clear that this is an observation made after the fact about the distribution of loss/cross-entropy, which is reflected in the way points are sampled. Currently, it reads as a part of the method, which is a bit confusing.

> we find that randomly selecting a subset of the data at each checkpoint is more effective

Again this suggests that the dataset has too much data? Unless this randomness is only within a specific subset of samples (which should be said explicitly here).

> Given this grouping of the dataset

Still not clear by this point in the paper how these grouping are obtained

> the training trajectory of a specific model does not factor into the
final pruned dataset.

That's not entirely true, in SVP and FS the trajectory of the proxy/initial model factors in--indirectly. One could imagine it's like if at the midpoint of training all the parameters were reset, but the training trajectory retains the knowledge of the first half of training to prune data.

> At each checkpoint, an algorithm may observe the current performance
of the model on the full dataset

That sounds expensive? Unless the training performance of the most recent epoch is cached and considered "current"? (from the EMA s_uncertainy explanation that follows, it seems like the answer is no? there's a full forward pass being made?)

> Uncertainty sampling

The word uncertainty has a fairly specific meaning in ML/statistics. I'm not sure it is correct to characterize categorial cross-entropy as a measure of uncertainty, since it is a measure of similarity between two distributions.

> So far, our decision making criterion has been fairly simple

At this point in the paper, no decision making process has been described. What is the criterion used for? Are all the samples ranked and the top k chosen? Are they sampled stochastically with the score being the unnormalized probability?

> multi-armed bandit problem from reinforcement learning

Comparing this selection process to a multi-armed bandit is a bit clumsy. The greedy decision here would be to select the _single sample_ with the highest loss at every iteration (assuming one arm == one data sample); this would be bad, and is not what is done here. An alternative could be to have two arms, one to select from the high-loss samples, one from the rest, and to make an $\epsilon$-greedy decision--although that breaks the analogy between value of arms and individual losses of data points.

If this were me, I'd either spend some time writing this down formally to fit the bandit/MDP framework, or just avoid making an innacurate comparison entirely. It's totally fair to say something along the lines of "we sort all data points, take the top k, and sample uniformly at random from the rest."

> The upper-confidence bound (UCB) (Sutton & Barto, 2018) algorithm is another RL method that selects arms based on the mean value and the variance

I'm not sure what is being referred to here. This is certainly not the UCB algorithm as described in Sutton & Barton (2018) section 2.7, which suggests that actions be taken such that

$$A_t = \arg\max_a\left[Q_t(a) + \sqrt{\frac{\ln(t)}{N_t(a)}} \right] $$

On the other hand, Eq (4) is exactly the same as what is used in **Bayesian optimization**'s UCB method. See for example _Gaussian Process Optimization in the Bandit Setting: No Regret and Experimental Design_, Niranjan Srinivas, Andreas Krause, Sham M. Kakade, Matthias Seeger, 2010.

BO as an underlying framework here is much better suited than RL. BO even has a number of papers on the concept of selecting _batches_ of candidates (which naturally applies here as I'm assuming the top k again are kept).


> Figure 2

There is a very important missing baseline here. If I understand correctly, training a model with pruning ratio $1-r$ for $N$ epochs on a dataset of $n$ points means that the model "sees" $rNn$ points in total. The "Full Baseline" sees $Nn$ points in total. There should be a line for models that are trained with dynamic pruning for $Nn$ points in total (i.e. for $N/r$ epochs). Would that line reach parity with the Full Baseline after having seen $Nn$ points? Before that? Such an experiment would help disentangle whether we are just seeing the effects of (too) early stopping or whether this method is doing something signficicant.

Section 4.8 kind of addresses this but in a roundabout way. Comparing to the full baseline trained for 100 epochs is the same as comparing to the random sampling with $r=50$% trained for 200 epochs.

Here's a suggestion: Make a plot where the x axis is the number of data points seen, and the y axis is the test accuracy (i.e. the typical training curve). Train models until they have seen $Nn$ points or have converged. For every method & $r$ value plot the curves. If they overlap, then what we're observing is essentially early stopping, and if they don't overlap (e.g. the test accuracy grows faster/slower for the proposed method than for the Full Baseline), then what we're observing is a non-trivial side effect of choosing different subsets of training points.

> Figure 3

The figure looks fine, but something eludes me. If the model is trained for 200 epochs with sparsity 70%, shouldn't the right limit of the x axis be 200*0.3=60?

>  even randomly sampling these points (as the random dynamic method does) is better. & Table 1 results

Isn't this unsurprising? By taking random samples from $n$ points rather than $rn$ points, the model sees more unique points. It is well known (see work on double descent [1,2,3]) that training a DNN on more data makes it have a lower test error, for equal training time.

[1] Reconciling modern machine learning practice and the bias-variance trade-off, Mikhail Belkin, Daniel Hsu, Siyuan Ma, Soumik Mandal
[2] High-dimensional dynamics of generalization error in neural networks, Madhu S. Advani, Andrew M. Saxe
[3] The jamming transition as a paradigm to understand the loss landscape of deep neural networks, Mario Geiger, Stefano Spigler, Stéphane d'Ascoli, Levent Sagun, Marco Baity-Jesi, Giulio Biroli, Matthieu Wyart

> The overall conclusion is that the remaining hyper-parameters have minimal effect on the end result.

This should raise an eyebrow. If the main knobs to control a method have no impact, then what are those hyperparameters changing? It should at least be possible to break the method by turning the knobs too far.

--

It does feel like there could be more baselines in this paper (which, I know, there can always be more of -- would a review really be a review if it didn't complain that baseline X is missing?). In particular, there are no comparisons to other dynamic methods (some of which were not designed for this purpose, but should be are more "interesting" than $\epsilon$-greedy and UCB):
[4] Self-Paced Learning for Latent Variable Models, M. Pawan Kumar, Benjamin Packer, Daphne Koller, 2010
[5] Online Batch Selection for Faster Training of Neural Networks, Ilya Loshchilov, Frank Hutter, 2015
[6] Prioritized Experience Replay, Tom Schaul, John Quan, Ioannis Antonoglou, David Silver, 2015
[7] Not all samples are created equal: Deep learning with importance sampling, Angelos Katharopoulos, François Fleuret, 2018
[8] BatchBALD: Efficient and Diverse Batch Acquisition for Deep Bayesian Active Learning, Andreas Kirsch, Joost van Amersfoort, Yarin Gal, 2019


**Strengths And Weaknesses:**

This paper is fairly well written and structured. It exposes the current state of methods in data selection, points out a problem, and proposes some solutions to it. It then experiments with the proposed solutions and tests some hypotheses as to why the method performs the way it does.

I do have several concerns about the paper:
- It does seem like some part of the literature is ignored. This paper positions itself in opposition to "static data pruning methods", but there exists works that approach this problem dynamically [4,5,7]; the proposed method is not the first one to do so
- The method is presented as an RL method, but it hardly is one---unless we are willing to stretch definitions quite a bit. In my opinion the paper would benefit from not getting RL involved at all here.
- The experimental results do not seem to provide clear evidence of the claims of the authors
    -  The experiments do not seem to disentangle between the effects of the method and the impact of early stopping/training on less data in total. The authors themselves remark the "unreasonable effectiveness of random pruning".
    -  I am not confident that the improvements made by the method are statistically significant. No confidence is reported, and visually, gains between random pruning and the UCB method are marginal---and considering the number of seeds, could be due to luck.

---

> ### Author Response · Authors · 2022-09-29
> **Initial response**
>
> Thank you for your review. Below are some specific comments to the points that you have raised.
>
> > Improving experiments to make sure that the observed results are consistent, and are not caused by simply seeing less data.
>
> > Again this suggests that the dataset has too much data? [...]
>
> > There is a very important missing baseline here. [...]
>
> This is an interesting point, and we have run experiments to generate the suggested "samples seen" plot (Fig. 5, revised). We ask the reviewer to read the surrounding discussion in Sec. 4.3 which gives a detailed interpretation of the plot. Our high-level conclusions are that the full baseline should act as an upper bound on the convergence w.r.t. samples seen. To do better than the full baseline, one would suppose there exists a set of samples that is "more instructive" of the true decision boundary. If such a set exists, then only training on those samples would outperform the baseline for a fixed samples seen budget. This is exactly the assumption that underlies the static pruning work, and our work suggests that this is a faulty assumption. The results in the plot further corroborate this finding, and they additionally illustrate the differences between the various methods proposed.
>
> > Fig 1.; it wasn't immediately clear the the $x$ indices changed from row to row [...]
>
> This is a good suggestion, and we will update the figure accordingly for the final version.
>
> > That's not entirely true, in SVP and FS [...]
>
> We have update the text to clarify that the trajectories only have an indirect influence.
>
> > That sounds expensive? [...]
>
> Yes, at each checkpoint, a full forward (but not backward) pass is done over the entire dataset. This overhead is included in the wall clock time plots, and as the figures show, it is not significant. Note that checkpointing is not done every epoch, but at a fixed pruning period which is a hyper-parameter of the decision making process. We choose a period of 10 epochs for our experiments.
>
> > The word uncertainty has a fairly specific meaning in ML/statistics. [...]
>
> We are specifically using the definition in Settles, 2009 (cited in the paper) and for multi-label classification, we believe that the cross-entropy meets their definition. But we would be willing to use a better descriptor if suggested.
>
> > Comparing this selection process to a multi-armed bandit is a bit clumsy. [...]
>
> While thinking about the problem loosely within the RL-framework guided our work in this paper, we agree with the reviewer that the analogy is weak. More importantly, it does not give additional insight into the problem or its possible solutions. For this reason, we have chosen to revise the paper to describe the problem without referencing RL as suggested.
>
> > I'm not sure what is being referred to here. This is certainly not the UCB algorithm as described in Sutton & Barton (2018) section 2.7 [...]
>
> Thank you for this correction. We have updated the citation in the text.
>
> > The figure looks fine, but something eludes me. [...]
>
> The x-axis is the number of times each sample is selected. For 200 epochs with a pruning period of 10 epochs, any sample can be selected at most 200 / 10 = 20 times.
>
> > Isn't this unsurprising? By taking random samples from $n$ points rather than $rn$ points, the model sees more unique points. [...]
>
> In general, we agree that this should be expected. More data is usually better, but this is because many samples will be sometimes samples. On the other hand, seeing more never samples provides no value. Moreover, if sometimes samples were clearly ordered, then randomly sampling them would underperform statically selecting the top $k$. This is the point being made in this portion of the text.
>
> > This should raise an eyebrow. If the main knobs to control a method have no impact, [...]
>
> Within reasonable ranges, there is minimal effect. Of course, we can push their values to a point where the methods break down.
>
> 1 / 2

---

> > ### Author Response · Authors · 2022-09-29
> > **Continued response**
> >
> > > Discuss additional related literature, ideally comparing to it experimentally.
> >
> > > It does feel like there could be more baselines in this paper [...]
> >
> > We thank the reviewer for brining the following works to our attention. Below is our initial review of suggested literature, and we have added a complete discussion to the related work section in the revised paper.
> >
> > Self-Paced Learning [4] (SPL) is concerned with the ordering of examples so that the training loss of the considered SVM is lowered. This method is related to curriculum learning which suggests for the model to learn on easy examples and then more difficult ones. More concretely, SPL begins by starting from a small subset of samples and increasing the budget during training by annealing a hyperparameter. The main distinction between SPL and our methods is that our motivation is focused on reducing wall-clock time for a given accuracy and SPL requires an initial set of parameters exposed to the entire dataset while our approaches only operate on subsets of the training data.
> >
> > Online Batch Selection (OBS) [5] is the most similar to our method by varying both the batch size as well as the the data samples chosen for training. They rely on using the objective function to rank examples and rescore the $r_{ratio}*N$ ($N$ is the total number of samples) based on a frequency parameter. These loss values could potentially be stale unlike in our method where we pause training the rescore all the samples. At each epoch, they compute a selection pressure parameter which dictates how large of subset should be used. Two difference between our work and theirs is that we have the same sample budget on every epoch whereas theirs potentially changes and our scoring methods rely on smoothing techniques like EMA (exponential moving averages), eps-greedy, and UCB. While the prior work relied on the sometimes samples, our work is the first to have clarified this phenomenon.
> >
> > Prioritized Experience Replay [6] (PRE) is similar to OBS in that certain transitions in RL training are replayed more than others based on a computed probability of being sampled. As in OBS, only the evaluated experiences are updated in the dictionary and they incorporate some level of stochasticity into the scoring mechanism by perturbing some of the probability of sampling a particular transition by small factor, epsilon. This method resembles our eps-greedy method but our method of scoring is based on the greedy approach, thresholding the sample budget and allocating the rest of budget to random sampling of the dataset. PRE is specific to RL, and it is not clear that there should be overlap with our work on sometimes samples.
> >
> > Importance Sampling [7] (IS) reduces the training time of deep learning by targeting samples which reduce the variance of stochastic gradients during training. They drive an upper-bound to the per-sample gradient norm which can be computed in a single forward pass of the data. After every step, they compute a quantity that checks whether a speedup can be achieved and if it reduces the gradient variance, a large subset is sampled from the entire dataset and samples are pulled from this larger pool as computing importance scores on the entire dataset is expensive. IS potentially has access to train on the entire dataset whereas our methods only train on subsets of the data. IS is more complex than our methods, and the effectiveness of random sampling suggests that increased complexity is unwarranted.
> >
> > BatchBALD [8] is a batch acquisition method for active learning which relies on using an approximation to the mutual information between a batch of data and the model parameters. This approach relies on a data point being included in the subset as it provides a high level of uncertainty for multiple realizations of the model parameters. The disadvantage to this method is that it is specific to Bayesian NNs and relies on MC sampling which would be very expensive to do in an online fashion since we are concerned with reducing wall-clock time. One other way to implement an approximation to BatchBALD would be to take the prior model checkpoints and pass the data through each model to obtain an average score for each sample and select a subset based on the sample budget; of course, this is an expensive operation and would need to be justified by increasing the pruning period.
> >
> > > I get the intended meaning, but "evaluating" can also mean running the validation. I'd just say "the per-iteration computational cost of the model".
> >
> > > Selected by which method? [...]
> >
> > > Still not clear by this point in the paper how these grouping are obtained
> >
> > > At this point in the paper, no decision making process has been described. [...]
> >
> > We have made these changes to the text.
> >
> > 2 / 2

---

> > ### Comment · Reviewer_Nr2k · 2022-10-03
> > **A few more questions**
> >
> > Thank you for your replies and additional Figure and explanations, they are helpful.
> >
> > I think I may have misread the _tone_ of the paper initially. Upon reading your rebuttal I am left with the impression that the goal of this paper is to be a _negative result_ paper. Is this correct?
> >
> > If this is the case, I would suggest making this more explicit in the paper.
> >
> > > the full baseline should act as an upper bound on the convergence w.r.t. samples seen
> >
> > This is an interesting assumption, doesn't it go contrary to previous literature on things like curriculum learning? It would be surprising if there were _no_ way to select or order samples in such a way that is faster than iid sampling.
> >
> > Here's another thing which makes me see this work as a negative result. In the abstract it is claimed that the proposed methods "can reduce the training time by up to 10× without significant performance loss". This doesn't seem like a fair statement; as per Figure 5, the full baseline when trained on 10x less data upper bounds all the proposed methods. Presumably training iid on 10x less data is just as fast if not faster than the proposed methods trained with the corresponding pruning ratios. While it is technically true that the proposed methods do not incur a significant performance loss, I fail to see what benefit they provide. It's extra code, extra hyperparameters to keep around, more complexity and confounding factors.
> >
> > As far as I know, static pruning methods do have in mind restricted settings where having a smaller dataset is considered valuable. For example Paul et al. claim as a motivation that:
> > > Memory and resource constrained settings, such as on-device computing, require smaller models and datasets
> >
> > although the paper presents itself more as a theoretical contribution than a practical one.
> >
> > The analysis of samples that is performed in this paper is interesting, as far as I know novel, and I think a pretty good insight into why static pruning methods may not be ideal.
> >
> > What is the proposed setting here? One still needs to keep the entire dataset around, since any part of it might be required, so memory usage is not addressed by the proposed methods. Performance is upper bounded by iid training, so there are no gains on this front. Since performance is upper bounded in time as well, regardless of how little compute access we have, it always seems better to use iid.
> >
> > Therefore, I'm a bit uneasy with statements such as
> > > In lieu of these improvements, our methods bring practical, efficient data pruning to DL researchers
> >
> > and
> > > By pruning the data, we allow models to be trained with fewer GPUs or in less time.
> >
> > Why would a DL researcher or practitioner use the proposed method rather than training on the full dataset?
> >
> > >  If such a set exists, then only training on those samples would outperform the baseline for a fixed samples seen budget. This is exactly the assumption that underlies the static pruning work, and our work suggests that this is a faulty assumption. The results in the plot further corroborate this finding, and they additionally illustrate the differences between the various methods proposed.
> >
> > It's great to point out faulty assumptions of past work, I'm all for that, but this isn't currently quite how the paper is being presented. As I've started this reply with, I think it makes more sense to view this paper as a negative result and/or as an insight into limitations of static methods. This would require some rewriting which for example removes claims of improved efficiency.
> >
> > Happy to hear your thoughts on this.

---

> > > ### Author Response · Authors · 2022-10-04
> > > **Response #2**
> > >
> > > > I am left with the impression that the goal of this paper is to be a _negative result_ paper. Is this correct?
> > >
> > > It depends what you mean by a negative result. If you mean that the goal of the paper is to invalidate the prior work, then this is not our intention. The prior work is still correct; it just fails to extract the full opportunity based on the analysis that we provide. And there are still cases where the prior work is applicable such as the memory constrained devices you mentioned (thank you for mentioning this detail---it should be included in our paper). But our work does call into question the assumptions built into the prior work and its wide spread utility.
> > >
> > > On the other hand, we agree that the central and most interesting result of our paper is the analysis of different sample groups. We have always wanted this finding to be the focus of the paper.
> > >
> > > We tried to find a case where reducing epochs vs. pruning samples is not equivalent. This would be the case where the number of epochs is small for the full baseline. Even though the full baseline would see all the same samples as a dynamic pruning method, it would not see those samples as often. Clearly, above some threshold, this does not impact performance, but if the total epochs is low enough such that each sample is seen too few times, then performance might suffer. This is even more true when there are never samples in the data, which is a wasted allocation of the samples seen budget. Instead of running the pruning methods for $T / k$ epochs like we do in Fig. 5, we repeat the experiment by running the full baseline for $kT$ epochs. _We find that our methods converge faster than the full baseline and to a higher final accuracy_ (with the UCB performing the best). Despite this showing that early stopping and data pruning are subtly different, the margin of difference is only 0.5% on CIFAR-10 at 80% pruning, so we do not think this is a compelling reason for data pruning. Our initial intent was not present a null result (if this is what you meant), but given the recent results of this review period, we agree that claims of improved efficiency should be removed.
> > >
> > > As you suggest, it is hard to believe there is no subset of samples that improves on the full baseline. The key difference between curriculum learning and data pruning is finding _easy_ vs. _informative_ samples. These are not necessarily equivalent, but our paper suggests that merging these two fields might be the most promising avenue for future data pruning work.

---

### Decision · Action_Editors · 2022-10-21

**Recommendation:** Reject

**Comment:**

Based on the criticisms outlined under "claims and evidence", the consensus was to reject the paper in its current form.

**Audience:**

Anyone who trains a machine learning model would be interested in methods for getting rid of data to accelerate training and converge to a better solution. Such methods are only useful if they save wall-clock time. The general setting of this paper is therefore of interest to a broad audience.

**Claims And Evidence:**

This paper focuses on "dynamic data pruning", an approach to filtering out samples that changes over the course of training (in opposition to static data pruning approaches that perform an expensive up-front step to choose which samples to train on). The central arguments of the paper are that 1) when accounting for the cost of the static data pruning step, existing methods don't decrease overall training time; 2) simple uniform pruning over the course of training actually outperforms prior methods with aggressive pruning rates; and 3) two newly-proposed dynamic data pruning methods can further improve performance. Reviewers generally believed the first and second claim, there was some disagreement that the third claim was convincingly made since the proposed pruning methods did not significantly outperform uniform pruning. Reviewers also felt that the new methods were proposed in a somewhat unclear way and that it was hard to tell whether the paper was aiming to make a negative point about existing pruning methods or a positive point about the newly proposed ones. Finally, one reviewer pointed out that there were missing baselines/prior work on dynamic data pruning; this could likely be fixed in a revised version of the paper.